# ASTrA: Adversarial Self-supervised Training with Adaptive-Attacks

**Prakash Chandra Chhipa**[1*], **Gautam Vashishtha**[2*], **Settur Anantha Sai Jithamanyu**[3*],
**Rajkumar Saini**[1], **Mubarak Shah**[4], **and Marcus Liwicki**[1]
[1]Luleå Tekniska Universitet, Sweden, [2]Indian Institute of Technology, Gandhinagar, India
[3]Indian Institute of Technology, Madras, India, [4]University of Central Florida, USA
[1]{prakash.chandra.chhipa, rajkumar.saini, marcus.liwicki}@ltu.se
[2]gautam.pv@alumni.iitgn.ac.in, [3]ed20b058@smail.iitm.ac.in
[4]shah@crcv.ucf.edu

## Abstract

Existing self-supervised adversarial training (self-AT) methods rely on hand-crafted adversarial attack strategies for PGD attacks, which fail to adapt to the evolving learning dynamics of the model and do not account for instance-specific characteristics of images. This results in sub-optimal adversarial robustness and limits the alignment between clean and adversarial data distributions. To address this, we propose *ASTrA* (**A**dversarial **S**elf-supervised **Tr**aining with **A**daptive-Attacks), a novel framework introducing a learnable, self-supervised attack strategy network that autonomously discovers optimal attack parameters through exploration-exploitation in a single training episode. ASTrA leverages a reward mechanism based on contrastive loss, optimized with REINFORCE, enabling adaptive attack strategies without labeled data or additional hyperparameters. We further introduce a mixed contrastive objective to align the distribution of clean and adversarial examples in representation space. ASTrA achieves state-of-the-art results on CIFAR10, CIFAR100, and STL10 while integrating seamlessly as a plug-and-play module for other self-AT methods. ASTrA shows scalability to larger datasets, demonstrates strong semi-supervised performance, and is resilient to robust overfitting, backed by explainability analysis on optimal attack strategies. Project page for source code and other details at https://prakashchhipa.github.io/projects/ASTrA.

## 1 Introduction

In an era where Convolutional Neural Networks (CNNs) are increasingly deployed in a wide range of critical applications across domains like medical image analysis (Wang et al. (2019); Ma et al. (2021); Kaviani et al. (2022)), object detection (Wang et al. (2021a); Hoory et al. (2020); Wei et al. (2018)), facial recognition (Vakhshiteh et al. (2020); Akhtar et al. (2021); Biswas et al. (2021)), autonomous driving (Cao et al. (2019); Sun et al. (2020); Tu et al. (2020)) among others, their susceptibility to adversarial attacks has become a pressing concern (Szegedy (2013); Hendrycks & Dietterich (2019)). These attacks, often imperceptible to human observers are referred to as Adversarial Examples (AEs), can cause deep models to fail catastrophically, undermining trust in AI systems.

Adversarial Training (AT) emerges as the most prominent defense against adversarial attacks in supervised learning. This approach injects Adversarial Examples (AEs) as part of the training regime of Deep Neural Networks (DNNs). By exposing DNNs to both standard and perturbed samples during training, AT improves generalization to adversarial inputs within a specified $\epsilon$-ball in the input space, resulting in increased invariance to perturbations Madry (2017).

Although existing works in supervised Adversarial Training (sup-AT) have shown improvements in adversarial robustness (Madry (2017); Zhang et al. (2019); Wang et al. (2021b)), their dependence on true class labels for crafting adversarial examples (AEs) limits their broader applicability. Most sup-AT methods utilize hand-crafted attack strategies, such as the Projected Gradient Descent (PGD)

---

*equal contribution

attack (Mkadry et al. (2017)), with predefined parameters like a maximal perturbation of 8, 10 iterations, and a step size of 2.

The past few years have seen significant progress in self-supervised learning (SSL) (Misra & Maaten (2020); Chen et al. (2020); Noroozi & Favaro (2016); Gidaris et al. (2018)), more specifically Contrastive Learning (CL) for learning representations without the need for ground truth (GT) labels. Prominent SSL methods, such as SimCLR (Chen et al. (2020)), learn robust representations by enforcing instance discrimination, treating each instance and its views as a separate class (Purushwalkam & Gupta (2020)).

Inspired by this progress, multiple recent works (Kim et al. (2020); Jiang et al. (2020); Fan et al. (2021); Luo et al. (2023)) have positively attempted to leverage unlabelled data for achieving adversarial robustness (self-AT). Built as a min-max optimization strategy in Adversarial Contrastive Learning, an 'attacker' crafts input perturbations that attempt to minimize representation similarity for worst-case robustness, and the target network, i.e 'defender', targets maximizing the representation similarity for improved robustness against such perturbed adversarial attacks.

ACL (Jiang et al. (2020)) integrates continual learning with adversarial training to establish a self-AT framework. Similarly, RoCL (Kim et al. (2020)) enhances adversarial CL by aligning the distributions of clean and perturbed images. AdvCL (Fan et al. (2021)) employs knowledge distillation using pseudo labels from pretrained self-supervised models. DeACL (Zhang et al. (2022)) introduces a two-stage approach that distills a standard pretrained encoder through adversarial training. Recently, DynACL (Luo et al. (2023)) investigated the impact of augmentation strength on adversarial pretraining.

While these existing self-AT methods made significant progress in improving robustness, they enhance representation learning without considering the impact of attack strategies on the learning dynamics. They rely on heuristic, domain-specific techniques to craft adversarial examples, using attack strategies derived from supervised adversarial training (sup-AT) heuristics. This limits their ability to dynamically adapt attack strategies, potentially compromising the robustness and effectiveness of the learned representations. In contrastive learning, representations are uniformly distributed in the feature space, adhering to the principle of maximum entropy (Jaynes (1957)) and typically residing on an $n$-dimensional hyper-sphere (Ermolov et al. (2021); Gupta et al. (2023)). Since most instances are positioned near class boundaries with dispersed class clusters, effective attack strategies should leverage knowledge of the learned representations to perturb samples minimally toward boundaries, thereby avoiding class confusion. This approach is not followed by any of the existing methods.

Although some studies have explored the effects of varying and adaptive attack strategies in sup-AT (Tramer et al. (2020); Yao et al. (2021); Jia et al. (2022)), it is crucial to investigate these impacts in the context of self-AT. This is because, unlike sup-AT, self-AT does not utilize ground truth labels, meaning that insights and methods from sup-AT cannot be directly applied. In sup-AT, attack strategies often rely on label information to craft targeted adversarial examples, optimize perturbations based on class-specific gradients, and evaluate attack success using label-based metrics. Without access to such labels, self-AT must generate and assess adversarial examples based solely on the data and learned representations. This leads to the following research questions:

1) What is the impact of employing different attack strategies at various training stages in self-AT?

2) How does sample-level variation in attack strategy selection affect robust representation learning in self-AT?

To answer these questions, our preliminary investigation results, shown in Figure 1, demonstrate that varying attack strategies both across training stages (left) and at the sample level (right) can enhance adversarial robustness. Specifically, implementing different attack schedules during various training phases leads to modest improvements over the baseline ACL. Additionally, introducing sample-level variation through random perturbations slightly outperforms the baseline ACL, highlighting the benefits of adaptive attack strategies. This suggests that samples have varying degrees of vulnerability to adversarial perturbations, and a one-size-fits-all attack strategy may not effectively challenge the model across all instances. However, these experiments still rely heavily on handcrafted heuristics and domain-specific knowledge, which limits their generalizability. As training progresses, the most effective adversarial examples for enhancing robustness may change, neces-

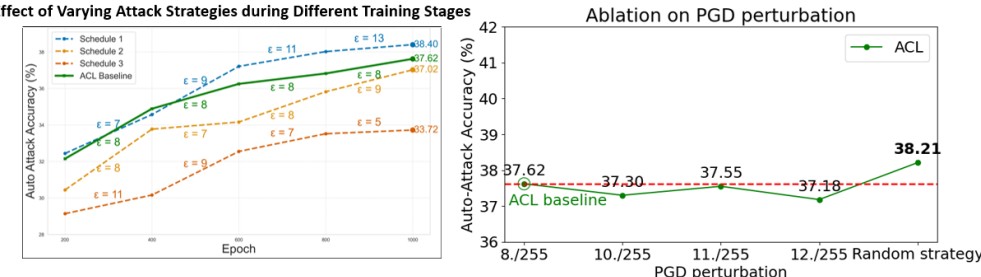

Figure 1: Ablation on PGD perturbations in ACL method. (left): Varying attack strategy by changing perturbations at different steps following different schedules. (right): Experiments with different perturbation values in handcrafted strategy followed by Random strategy where sample-level perturbations are chosen randomly from range 3 to 14.

sitating adaptive strategies that respond to the model's evolving performance. Moreover, existing instance-level variations do not account for sample characteristics, resulting in attack strategies that are not truly sample-dependent.

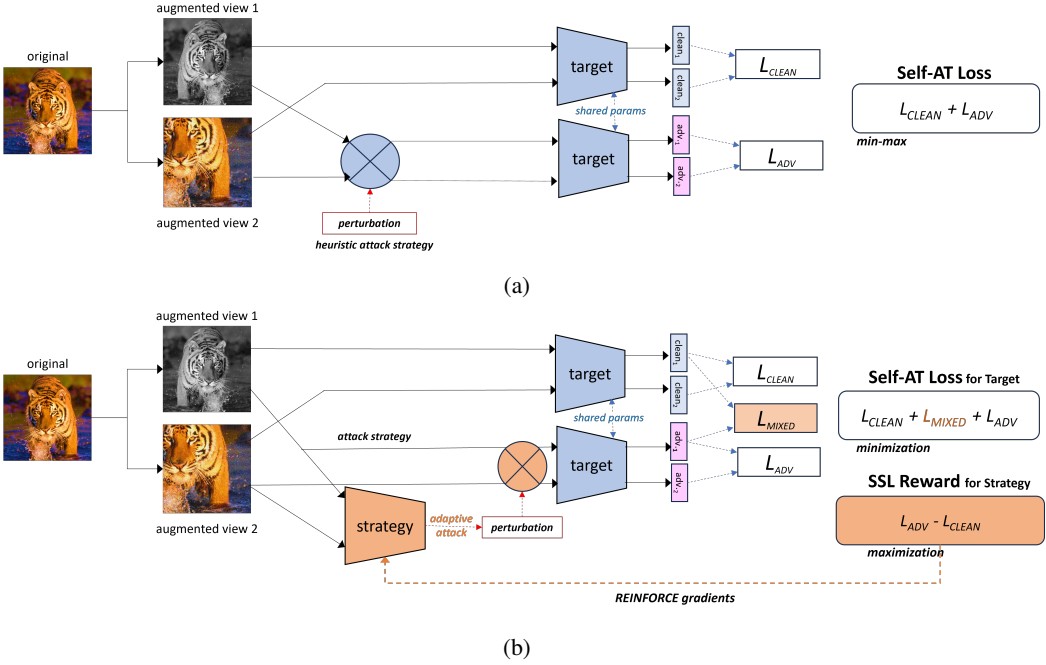

Figure 2: (a) Conventional self-AT with heuristics PGD attack. Here, samples are perturbed using handcrafted attack strategies causing limited scope of adversarial robustness. (b) ASTrA with learnable adaptive attack strategy and mixed contrastive objective. In this case, attack strategies are selected by the adaptive Strategy Network depending upon sample-characteristics and training dynamics of the target model.

To address these challenges, we propose Adversarial Self-supervised Training with Adaptable Attacks (ASTrA). ASTrA incorporates a novel learnable attack strategy module (refer Fig. 2b) that employs an exploration-exploitation approach (refer Fig. 3b) within a single episodic framework to identify optimal attack strategies for each training instance. This is achieved via a novel optimization formulation that leverages policy gradient methods that enable the strategy network to adjust attack strategies by directly influencing the target network's learning trajectory to maximize the robustness of the network. In the exploration phase, the strategy network explores a wide range of attack parameters to gather information about how different strategies affect the model's learning. Over time, it converges towards the most effective attack strategies based on feedback from the model's performance, refining the attack parameters to continually challenge the model appropriately, which can be seen in Fig. 3b. Our approach uniquely applies adaptive attacks during the self-supervised

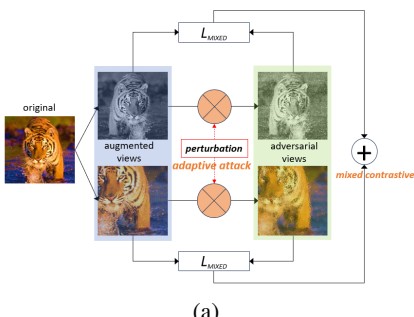
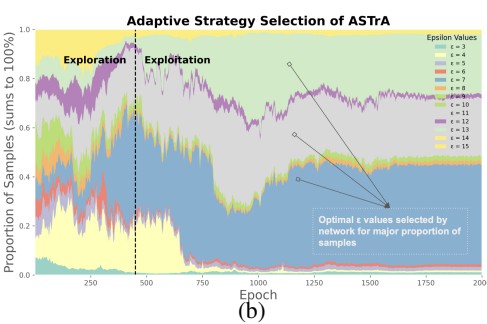

(a)
(b)

Figure 3: (a) ASTrA's mixed contrastive loss objective facilitating distribution alignment between clean and perturbed samples. Comparison with RoCL in sec. A.3 in appendix. (b) Exploration and Exploitation trends of ASTrA. The strategy model initially explores across perturbation values (exploration) and then assigns optimal values ($\epsilon = 7, 11, 13$) to maximal proportion of images (exploitation) as training progresses for better generalization.

pretraining phase, in contrast to adaptive sup-AT methods (Jia et al. (2022)) that utilize adaptability in a fully supervised context.

Additionally, we propose Mixed Contrastive objective (refer Fig. 3a) integrated in our framework to mitigate the misalignment between adversarial and standard data distributions. Refer sec. A.1 for details on strategy and target network interactions and sec. A.2 for ASTrA's algorithm.

Our framework outperforms existing self-AT methods in adversarial robustness across multiple datasets, including CIFAR10, CIFAR100, and STL100, and scales effectively to larger datasets such as ImageNet-100. Furthermore, ASTrA operates as a plug-and-play module, demonstrating enhanced performance over several state-of-the-art self-AT approaches (see Table 5).

Our main contributions are:

(1) Self-supervised Learnable Attack Strategy: We introduce a novel, self-supervised, adaptable adversarial attack strategy that eliminates the need for human supervision and heuristics. This learnable strategy optimizes attack strategies at the instance level to maximize adversarial robustness, whilst maintaining standard accuracy, surpassing the capabilities of conventional self-AT methods.

(2) Mixed Contrastive Objective: We propose a mixed contrastive objective to address the distribution alignment challenge between clean and perturbed sample representations, thereby enhancing generalization and robustness.

## 2 METHODOLOGY

### 2.1 PIPELINE OF THE PROPOSED FRAMEWORK

The proposed ASTrA framework integrates two main components: Self-supervised Target Network and Self-supervised Strategy Network. This framework addresses the challenges outlined in the introduction by adaptively optimizing attack strategies and improving representation robustness against adversarial perturbations without relying on handcrafted heuristics or ground truth labels (refer Fig. 2b).

**Self-supervised Target Network.** Denoted as $\hat{y} = f_w(x)$, where $w$ represents target network parameters, this network learns robust feature representations by processing dual input streams: clean augmented views and adversarially perturbed counterparts following ACL Jiang et al. (2020). Standard augmentation techniques (e.g., cropping, rotation, jitter) are applied to clean images, while the adversarial perturbations are generated based on strategies learned by the Strategy Network.

**Self-supervised Strategy Network.** We introduce a novel, learnable, self-supervised strategy network designed to overcome the limitations of fixed attack strategies employed by existing self-AT methods. Unlike handcrafted strategies, the strategy network curates sample-specific attacks considering both the sample characteristics and the training dynamics of the target model to generate optimal attacks that best improve robust generalization (refer Fig. 2b). Self-supervision drives attack strategy selection through contrastive rewards. Initially, the strategy network explores a wide range of attack strategies (exploration), and as training progresses, it shifts towards exploiting the

learned dynamics and using the most suitable attack strategies (exploitation). This entire process occurs within a single training episode, as shown in Figure 6 (row 1). Formally, the strategy network generates an attack strategy $a = \{a_1, a_2, ..., a_M\} \in A$, where each $a_m$ is a discrete attack parameter (e.g., PGD step size $\alpha$, number of iterations $I$, perturbation strength $\epsilon$). These parameters are drawn from a conditional distribution $p(a|x; \theta)$, parameterized by $\theta$, and updated in real-time based on the feedback loop created by the target network's performance. ASTrA addresses the gradient-based updating of the Strategy Network's parameters by utilizing the **REINFORCE** algorithm Williams (1992). This allows us to bypass the non-differentiability of the adversarial example generation process and optimize the strategy network's attack policies through policy gradients. Details of this optimization are discussed in Section 2.4.

**Adversarial Example Generator.** The adversarial example generation process is central to the interaction between the two networks. For a given input $x$, an adversarial example $x_{\text{adv}}$ is crafted as:

$$x_{\text{adv}} = x + \delta \quad \text{where} \quad \delta = g(x, a(\theta), w), \tag{1}$$

where $\delta$ represents the adversarial perturbation and $g(\cdot)$ encapsulates the perturbation method (typically PGD). Conventional self-supervised adversarial training methods (ACL Jiang et al. (2020)), utilize fixed strategies, $a = a_{fixed}$, for generating adversarial examples. The adversarial examples challenge the Target Network to improve its robustness. The attack strategies are sampled from a probability distribution $p(a|x; \theta)$, where $\theta$ is updated during the training.

## 2.2 THE REWARD FOR THE STRATEGY NETWORK

The reward function for the Strategy Network is designed to balance adversarial robustness with feature consistency across clean examples, leveraging contrastive loss terms that are inherently self-supervised and label-free. This reward mechanism allows the Strategy Network to craft adaptive attack strategies that optimize the perturbation without access to ground truth labels while ensuring the model generalizes well on the clean data distribution.

**Adversarial Contrastive Loss:** The adversarial contrastive loss is structured to measure the dissimilarity between features of adversarially perturbed augmented views of the same image. By maximizing this dissimilarity, the Strategy Network is encouraged to explore adversarial strategies that create stronger perturbations, thereby enhancing the model's robustness against adversarial attacks:

$$L_{\text{adv}}(w, \theta) = -\log \frac{\exp(\text{sim}(f_w(x_i^{adv}), f_w(x_j^{adv}))/\tau)}{\sum_{k \neq i} \exp(\text{sim}(f_w(x_i^{adv}), f_w(x_k^{adv}))/\tau)}, \tag{2}$$

where $x_i^{adv}$ and $x_j^{adv}$ depend upon $\theta$ as per 1 and $x_k^{adv}$ represents all adversarial samples in the batch excluding $x_i^{adv}$. The temperature parameter $\tau$ controls the sensitivity of the softmax distribution.

**Clean Contrastive Loss:** Conversely, the clean contrastive loss minimizes the distance between features of unperturbed augmented views. This ensures that essential information from non-adversarial data is preserved, allowing the model to maintain high performance on the clean data distribution:

$$L_{\text{clean}}(w) = -\log \frac{\exp(\text{sim}(f_w(x_i^{clean}), f_w(x_j^{clean}))/\tau)}{\sum_{k \neq i} \exp(\text{sim}(f_w(x_i^{clean}), f_w(x_k^{clean}))/\tau)}, \tag{3}$$

where $x_i^{clean}$ and $x_j^{clean}$ are unperturbed augmented views of $x$, and $x_k^{clean}$ includes all clean samples in the batch excluding $x_i^{clean}$. By minimizing this loss, the Strategy Network ensures that perturbations do not overly distort the learned representations, preserving the model's ability to generalize to standard clean inputs.

**Reward Objective:** The composite reward objective for the Strategy Network, aimed at striking a balance between robust and standard performance can be written as:

$$R_{strategy}(\theta, w^{fixed}) = \mathbb{E}_{x \sim D} \left[ \mathbb{E}_{a \sim p(a|x;\theta)} \left[ \alpha L_{adv}(\theta, w^{fixed}) - \gamma L_{clean}(w^{fixed}) \right] \right], \tag{4}$$

where $\alpha$ and $\gamma$ are hyperparameters that balance the trade-off ensuring feature consistency (through clean loss minimization) and enhancing adversarial robustness (through adversarial loss maximization).

The reward objective can adapt to sample-level variations by continuously learning from the evolving state of the Target Network during training. As the Target Network processes each sample, the Strategy Network adjusts its attack strategy based on the reward feedback. This dynamic interaction

enables the Strategy Network to generate instance-specific perturbations tailored to both the input data and the model's training progress, leading to more effective adversarial challenges without dependence on any predefined, heuristic-based attack schedules.

## 2.3 The Loss Terms for the Target Network

The Target Network is trained using a composite loss function that integrates three distinct contrastive losses: clean, adversarial, and mixed. Each loss plays a critical role in ensuring generalization on clean data while maintaining robustness against adversarial attacks.

**Clean and Adversarial Contrastive Losses:** The clean and adversarial contrastive losses are aligned with those in the Strategy Network (refer to eq. 3 and 2). The clean loss preserves the network's performance on benignly augmented views, while the adversarial loss minimizes the distance between adversarially perturbed views, enhancing robustness. Minimizing these losses ensures the network performs effectively on both clean and perturbed inputs.

**Mixed Contrastive Loss:** This novel proposed loss focuses on aligning the distribution of clean and adversarial samples in the representation space. This alignment prevents robust overfitting by ensuring generalization across a wide range of adversarial attacks:

$$L_{\text{mixed}}(w, \theta) = -\log \frac{\exp(\text{sim}(f_w(x_i^{clean}), f_w(x_i^{adv}))/\tau)}{\sum_{k \neq i} \exp(\text{sim}(f_w(x_i^{clean}), f_w(x_k^{adv}))/\tau)}. \quad (5)$$

**Composite Objective:** The overall objective can be written as:

$$L_{target}(\theta^{fixed}, w) = \mathbb{E}_{x \sim D}[\alpha L_{adv}(w, \theta^{fixed}) + \beta L_{mixed}(w, \theta^{fixed}) + \gamma L_{clean}(w)], \quad (6)$$

where $\alpha$, $\beta$, and $\gamma$ balance the contributions of each term. The mixed loss facilitates continuous alignment between clean and adversarial distributions, supporting the generation of adaptive adversarial strategies while maintaining robust generalization.

## 2.4 Novel Optimization Formulation of ASTrA using REINFORCE

The standard formulation of the adversarial training involves a min-max formulation of the objective function with adversarial samples generated using hand-crafted adversarial attacks.

$$\min_w \mathbb{E}_{x \sim \mathcal{D}} \left[ \mathcal{L}(x) + \lambda \cdot \max_{\delta \in \mathcal{S}} \mathcal{L}(x + \delta) \right].$$

In our framework, we use two separate objective functions, one for the optimization of the strategy network and one for the target network. The interaction between the two networks occurs through adversarial sample generation from the attack strategies generated by the strategy network. Given equation 4 and 6, we formulate the min-max optimization of ASTrA with for adversarial training as follows:

$$\min_w \max_\theta L_{target}(\theta^{fixed}, w) + R_{strategy}(\theta, w^{fixed}). \quad (7)$$

A key challenge in optimizing the Strategy Network comes from the non-differentiable nature of adversarial example generation, making traditional gradient-based methods ineffective. The selection of attack parameters like intensity and perturbation type involves non-differentiable operations that backpropagation cannot handle.

To alleviate this issue, we employ the REINFORCE algorithm Williams (1992), a policy gradient method that allows optimization without requiring differentiable operations. This method enables the Strategy Network to update its parameters based on rewards derived from the Target Network's response, ensuring dynamic learning of attack strategies without labels.

The objective function $J(\theta)$ for the Strategy Network is to maximize the expected reward, which evaluates the success of adversarial attacks:

$$J(\theta) = \mathbb{E}_{x \sim D} \left[ \sum_{a \sim p(a|x;\theta)} R(x, a; \theta) p(a|x; \theta) \right], \quad (8)$$

where $R(x, a; \theta)$ measures the effectiveness of generated adversarial examples. The REINFORCE algorithm estimates the gradient of $J(\theta)$ as:

$$\nabla_\theta J(\theta) = \mathbb{E}_{x \sim D} \left[ \sum_{a \sim p(a|x;\theta)} R(x, a; \theta) p(a \mid x; \theta) \nabla_\theta \log p(a \mid x; \theta) \right] \tag{9}$$

$$= \mathbb{E}_{x \sim D} \left[ \mathbb{E}_{a \sim p(a|x;\theta)} \left[ R(x, a; \theta) \nabla_\theta \log p(a \mid x; \theta) \right] \right].$$

This gradient approximation, based on sampled strategies, allows the Strategy Network to refine its attacks by updating parameters $\theta$ according to the observed rewards from the Target Network's performance.

**Gradient Ascent Update:** The Strategy Network updates its parameters via gradient ascent:

$$\theta_{t+1} = \theta_t + \eta_\theta \nabla_\theta J(\theta_t), \tag{12}$$

where $\eta_\theta$ is the learning rate. These updates iteratively refine the attack strategies by adjusting $\theta$ to increase adversarial success.

**Synchronization:** REINFORCE facilitates managing the non-differentiable updates between the Target and Strategy Networks. While the Strategy Network optimizes its attack strategies, the Target Network concurrently updates its parameters to minimize contrastive losses. Reward from the Target Network informs the Strategy Network's gradient updates, creating a synchronized, co-evolving system that adapts continuously in a self-supervised manner. Refer to section A.1 for more details.

## 3 EXPERIMENTS

We evaluate ASTrA on the benchmarks CIFAR10, CIFAR100 Krizhevsky et al. (2009), and STL10 Coates et al. (2011), comparing against existing self-AT methods: RoCL Kim et al. (2020), ACL Jiang et al. (2020), AdvCL Fan et al. (2021), DeACL Zhang et al. (2022), DYNACL Luo et al. (2023), and DYNACL-AIR Xu et al. (2024). Additionally, we assess the scalability of ASTrA on the ImageNet-100 Tian et al. (2020).

**Pretraining.** We use ResNet-18 He et al. (2016) as the target network, incorporating a mixed contrastive loss term with a weighting parameter $\gamma = 0.5$, following the protocol from Jiang et al. (2020). ResNet-18 is also used as the adaptive strategy network, with a learning rate of 0.1, LARS optimizer, step sizes ranging from 1 to 6, attack iterations between 3 and 14, and a perturbation range of 3 to 15. Reward weights $\alpha$ and $\beta$ are both set to 1.0 for adversarial and clean losses, respectively. In ResNet, we use bottleneck projector head of size 2048x512, performance comparison with ACL (Jiang et al. (2020) using ACL projector head is in sec. A.4.4 in appendix. We set $\beta$ to 0.5 for mixed contrastive loss term. ASTrA++ is longer pretraining variant of ASTrA with 2000 epochs.

**Evaluation.** The learned representations are evaluated using three protocols: standard linear finetuning (SLF), adversarial linear finetuning (ALF), and adversarial full finetuning (AFF) for three accuracy metrics - Auto Attack Accuracy (AA), accuracy under PGD-20 as Robust Accuracy (RA), and Standard Accuracy (SA). SLF and ALF freeze the encoder and tune the classifier using natural (SLF) or adversarial (ALF) samples with cross-entropy loss. For AFF, the pretrained encoder is used as initialization, and the entire model is trained, following the approach in ACL Jiang et al. (2020). ACL, RoCL, AdvCL, DeACL results are reported from DeACL (Zhang et al. (2022)).

Table 1: SLF results on CIFAR10, CIFAR100, and STL10. All the methods are evaluated with ResNet18 under the same condition following Jiang et al. (2020). For all metrics (AA, RA, SA), Top two performances highlighted in **bold**. ASTrA++ denotes longer pre-training for 2000 epochs.

| SSL-AT | CIFAR10 | | | CIFAR100 | | | STL10 | | |
|---|---|---|---|---|---|---|---|---|---|
| | AA | RA | SA | AA | RA | SA | AA | RA | SA |
| Supervised | 46.23 | 47.50 | 84.35 | 23.27 | 25.86 | 58.98 | 29.21 | 31.34 | 49.38 |
| RoCL | 26.12 | 28.40 | 77.90 | 8.72 | 11.52 | 42.93 | 26.51 | 28.21 | **78.19** |
| ACL | 37.62 | 40.02 | 79.32 | 15.68 | 17.10 | 45.34 | 33.24 | 35.62 | 71.21 |
| AdvCL | 37.46 | 40.54 | 73.23 | 15.45 | 17.05 | 37.58 | 45.26 | 46.18 | 72.11 |
| DeACL | 45.31 | **53.95** | 80.17 | 20.34 | **30.74** | 52.79 | 45.54 | 46.72 | 72.82 |
| DYNACL | 45.04 | 46.72 | 77.41 | 19.25 | 21.40 | 45.73 | 46.59 | 47.38 | 69.67 |
| DYNACL-AIR | 45.17 | - | 78.08 | 20.45 | - | 46.84 | 47.66 | - | 72.30 |
| ASTrA | **46.40** | **54.02** | **80.54** | **21.34** | 24.28 | **53.20** | **47.62** | **48.82** | 78.00 |
| ASTrA++ | **46.92** | 53.10 | **80.46** | **21.95** | 25.10 | 53.58 | **48.21** | **49.26** | **78.72** |

## 4 RESULTS AND ANALYSIS

We evaluate ASTrA across multiple benchmarks against existing self-AT and sup-AT methods. ASTrA outperforms prior approaches in robustness, scalability, and adaptability across various evaluation protocols and datasets.

### 4.1 ROBUSTNESS ON MULTIPLE BENCHMARKS

In Table 1, we compare the robustness of supervised AT and various self-AT methods on CIFAR10, CIFAR100, and STL10. ASTrA outperforms prior self-AT methods, improving AA accuracy over ACL Jiang et al. (2020) by 8.78% on CIFAR10 (to 46.40%), 5.66% on CIFAR100 (to 21.34%), and 2.36% on STL10 (to 47.62%). ASTrA++ further enhances robustness, adding 0.52% on CIFAR10, 0.61% on CIFAR100, and 0.59% on STL10. Both ASTrA and ASTrA++ surpass recent self-AT methods like DYNACL Luo et al. (2023) and DYNACL-AIR Xu et al. (2024), demonstrating generalization and scalability. Notably, ASTrA++ exceeds supervised vanilla AT on CIFAR10 and STL10, marking a significant advancement in self-AT. Sec. A.4.3 in the appendix shows the detailed sensitivity analysis of the strategy network's learning rate in ASTrA.

### 4.2 ROBUSTNESS ON DIFFERENT EVALUATION PROTOCOLS

In Table 1, 2, and 3, we evaluate ASTrA and ASTrA++ under different protocols: SLF, AFF, and ALF. ASTrA consistently demonstrates state-of-the-art robustness compared to other self-AT methods across all protocols. Under ALF, ASTrA++ surpasses existing self-AT methods, including DYNACL and DYNACL-AIR, with a notable AA improvement of 1.18% on CIFAR10. In the challenging AFF settings, ASTrA improves upon the sup-AT baseline by 1.88% (from 48.96% to 50.84%) and shows superior performance on CIFAR10 and STL10. Overall, ASTrA and ASTrA++ achieve state-of-the-art results across diverse protocols, confirming their robustness and flexibility.

Table 2: AFF results on CIFAR10, CIFAR100, and STL10. ++ denotes longer pre-training.

| SSL-AT | CIFAR10 | | | CIFAR100 | | | STL10 | | |
|---|---|---|---|---|---|---|---|---|---|
| | AA | RA | SA | AA | RA | SA | AA | RA | SA |
| Supervised | 48.96 | 49.90 | 80.23 | 22.16 | 26.38 | 53.34 | - | - | - |
| RoCL | 47.88 | 51.35 | 81.01 | 22.28 | 27.49 | 55.10 | 28.88 | 30.20 | **80.10** |
| ACL | 49.27 | 52.82 | 82.19 | 23.63 | 29.38 | 56.61 | 34.85 | 35.42 | 75.11 |
| AdvCL | 49.77 | 52.77 | 83.62 | 24.72 | 28.73 | 56.77 | 46.70 | 47.80 | 76.20 |
| DeACL | 50.39 | 54.18 | **83.95** | 25.48 | 29.65 | 59.86 | 47.35 | 48.24 | 77.30 |
| DYNACL | 50.54 | 54.26 | 81.94 | 25.05 | 29.10 | 59.30 | 48.12 | 49.85 | 73.75 |
| DYNACL-AIR | 50.60 | - | 82.14 | 25.34 | - | 57.44 | 48.10 | - | 73.10 |
| ASTrA | **50.84** | **54.90** | 82.68 | **26.10** | **30.22** | 59.92 | **49.65** | **52.40** | **80.20** |
| ASTrA++ | **51.20** | **55.01** | 83.72 | **26.45** | **31.00** | 60.25 | **50.15** | **53.28** | 79.70 |

### 4.3 ASTRA'S SCALABILITY AND ABLATIONS ON CONTRIBUTED COMPONENTS

In Table 4, we evaluate ASTrA's scalability on CIFAR10, CIFAR100, STL10, and the larger ImageNet-100 dataset using the AFF protocol. ASTrA effectively scales to larger datasets, achieving competitive performance on ImageNet-100 demonstrating adaptability to more complex tasks. We also conduct ablations on ASTrA's core components across all datasets. Including the mixed contrastive loss (MC) enhances performance, improving AA accuracy—for example, by 0.77% on CIFAR10. The adaptive attack strategy (A-Attack) yields further improvements, such as a 1.79% AA gain on CIFAR100.

Table 3: ALF results on CIFAR10.

| Pretraining Method | AA | RA | SA |
|---|---|---|---|
| Sup-AT | 47.00 | 48.12 | 83.22 |
| RoCL | 29.69 | 28.72 | 75.62 |
| ACL | 40.91 | 42.00 | 76.57 |
| AdvCL | 37.28 | 40.58 | 73.15 |
| DYNACL | 45.72 | 46.90 | 72.87 |
| DYNACL-AIR | 46.01 | - | 77.42 |
| ASTrA | **46.54** | **47.62** | **78.23** |
| ASTrA++ | **46.90** | **48.12** | **78.77** |

Combining both components achieves the highest gains, with AA improvements of 1.57% on CIFAR10 and 2.59% on CIFAR100. These results underscore the importance of both Mixed Contrastive and adaptive adversarial policies in enhancing ASTrA's robustness across datasets. The effect of batch frequency updates is detailed in sec. A.4.5. Computational overhead analysis on strategy network is in sec. A.4.1. Comparison of ASTrA++ with post processing variants of other methods is in sec. A.4.2.

Table 4: Ablations on the effect of each component of ASTrA. AFF results and scalability analysis. *MC:Mixed Contrastive, A-Attack: Adaptive Attack Strategy, A-Attack + MC: Complete ASTrA.*

| SSL-AT | CIFAR10 | | | CIFAR100 | | | STL10 | | | ImageNet-100 | | |
|---|---|---|---|---|---|---|---|---|---|---|---|---|
| | AA | RA | SA | AA | RA | SA | AA | RA | SA | AA | RA | SA |
| ACL | 49.27 | 52.82 | 82.19 | 23.63 | 29.38 | 56.61 | 34.85 | 35.42 | 75.11 | 09.24 | 11.51 | 19.22 |
| MC | 50.04 | 53.58 | 82.34 | 24.96 | 29.82 | 58.22 | 47.05 | 48.40 | 77.62 | 10.48 | 14.18 | 21.74 |
| A-Attack | 50.58 | 54.34 | 82.56 | 25.42 | 30.10 | 59.40 | 48.40 | 51.10 | 79.00 | 11.02 | 15.12 | 22.50 |
| A-Attack + MC | **50.84** | **54.90** | 82.68 | **26.10** | **30.22** | 59.92 | **49.65** | **52.40** | **80.20** | **11.21** | **15.38** | **23.01** |

### 4.4 ASTRA AS PLUG-N-PLAY FRAMEWORK

As shown in Table 5, integrating ASTrA's adaptive attack policy into RoCL and DYNACL enhances their robustness across all metrics. Specifically, ASTrA improves AA by 1.22% and RA by 1.89% when combined with RoCL, and boosts AA by 1.38% and RA by 0.78%

Table 5: ASTrA as plug-N-play framework with RoCL and DYNACL. AFF on CIFAR10.

| Metrics | RoCL | RoCL+ASTrA | DYNACL | DYNACL+ASTrA |
|---------|------|------------|--------|--------------|
| AA | 47.88 | **49.10** | 50.54 | **51.92** |
| RA | 51.35 | **53.24** | 54.26 | **55.04** |
| SA | 81.01 | **82.01** | 81.94 | **82.55** |

with DYNACL. These results demonstrate ASTrA's effectiveness as a plug-and-play framework.

## 4.5 ASTRA UNDER SEMI-SUPERVISED SETTINGS

As shown in Figure 4, ASTrA outperforms ACL in semi-supervised settings on CIFAR10, demonstrating clear improvements in Auto-Attack (AA), Robust Accuracy (RA), and Standard Accuracy (SA) across all label ratios. At a 50% label ratio, ASTrA achieves a significant 2.58% improvement in AA and a 1.78% gain in RA over ACL. Even with fewer labels, ASTrA maintains superior performance, highlighting its robustness with limited labeled data.

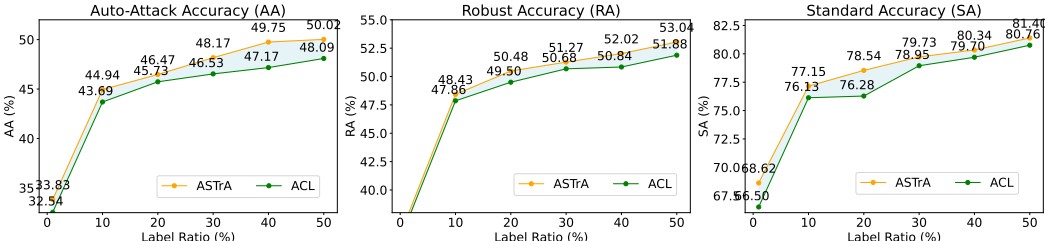

Figure 4: ASTrA consistently outperforms ACL in semi-supervised settings. AFF on on CIFAR10.

## 4.6 ASTRA ON ROBUST OVERFITTING

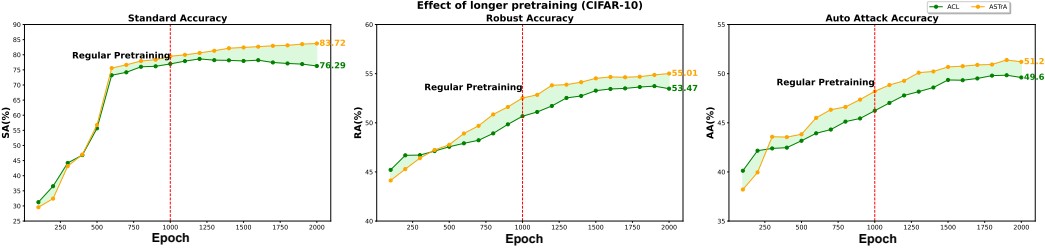

Figure 5: Longer AT - ACL vs ASTrA on CIFAR10. SA drops off for ACL.

ASTrA effectively mitigates robust overfitting during longer pretraining on CIFAR10, as shown in Figure 5. Over extended epochs, ASTrA maintains both Robust Accuracy (RA) and Standard Accuracy (SA), reaching RA of 55.01% and SA of 83.72% after 2000 epochs. In contrast, ACL experiences a significant decline in SA after 1000 epochs, indicating overfitting. This demonstrates ASTrA's superior ability to balance robustness and generalization during long-term training.

## 4.7 ASTRA ON FINDING OPTIMAL ATTACK PARAMETERS

ASTrA autonomously discovers optimal attack parameters, including step size, perturbation, and attack iterations, by employing a dynamic exploration-exploitation approach. Initially, the strategy network explores a wide range of values, searching for effective attack configurations. ASTrA shifts from exploration to exploitation as training progresses, converging on optimal values that balance adversarial robustness and generalization. This adaptive process in Figure 6 (row 1), is powered by the novel reward mechanism, which aligns model dynamics to fine-tune attack strategies. Our analysis across datasets, including CIFAR10, STL10, and ImageNet-100 (Fig. 6 (row 2)), further shows how ASTrA tailors its attack policies based on dataset-specific characteristics. Starting with broad exploration, ASTrA gradually narrows the range of perturbation values, finding optimal points for each dataset. This process requires no manual hyperparameter tuning, as ASTrA efficiently balances exploration and exploitation throughout training, ensuring robust defenses without sacrificing clean data performance. Ablations on discretization of the attack parameters is in sec. A.4.6.

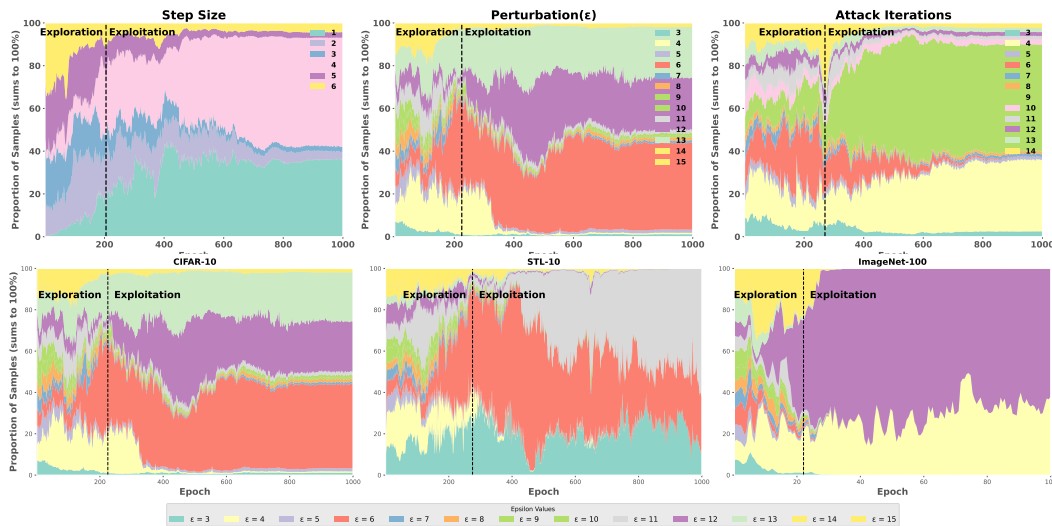

Figure 6: Exploration-exploitation phenomena of ASTrA. (row 1) ASTrA on CIFAR10 finding optimal attack policy (PGD iterations, perturbation($\epsilon$), step size) by exploration-exploitation phenomena by learnable self-supervised strategy module where entire training is a single episode (row 2) ASTrA finding optimal value(s) of perturbation by exploration-exploitation across three datasets - CIFAR10, STL10, and ImageNet100.

## 4.8 COMPARING OF ASTrA'S ADAPTIVE WITH RANDOM AND HANDCRAFTED ATTACKS

In the introduction, we questioned whether static or random strategies could adapt to the dynamic nature of adversarial training. While the random and scheduled strategies show slight improvements (Figure 7), they still fall short in fully optimizing attack parameters due to their lack of adaptability.

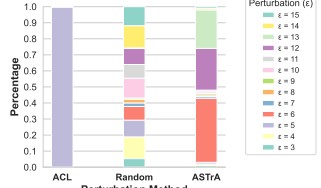

Figure 7: Perturbation strategies.

The random strategy lacks convergence, and the scheduled approach remains inflexible to the evolving model dynamics. However, these initial results indicated that an adaptive strategy could improve robustness by adjusting to these changing conditions.

Building on this insight, ASTrA's learnable adaptive strategy dynamically balances exploration and exploitation throughout training. As shown in Table 6, ASTrA surpasses both random and scheduled approaches, autonomously finding optimal attack parameters. The flatter loss landscape (refer sec. A.5) further demonstrates ASTrA's effectiveness in maintaining both adversarial robustness and generalization, validating the need for an adaptive strategy raised in the introduction.

Table 6: Comparison of attack strategies on SLF evaluation on CIFAR10.

| Strategy | AA(%) |
|---|---|
| ACL | 37.65 |
| Random | 38.21 |
| Scheduled | 38.40 |
| ASTrA (Adaptive) | **46.40** |

## 5 CONCLUSION

This work addressed the limitations of static adversarial strategies in self-AT by introducing *AS-TrA*, a self-supervised, adaptive attack framework. ASTrA autonomously optimizes attack parameters through a contrastive reward mechanism, using REINFORCE to enable dynamic, label-free adaptation. The framework significantly enhances adversarial robustness and generalization across benchmarks like CIFAR10, CIFAR100, and STL10, with scalability to larger datasets. Our analysis shows that adaptive, instance-specific strategies not only mitigate robustness overfitting but also provide deeper insights into finding optimal adversarial attack parameters, further strengthening the framework's ability to counter diverse adversarial threats.

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

# A APPENDIX

## A.1 INTERACTION BETWEEN STRATEGY AND TARGET NETWORKS IN ASTRA

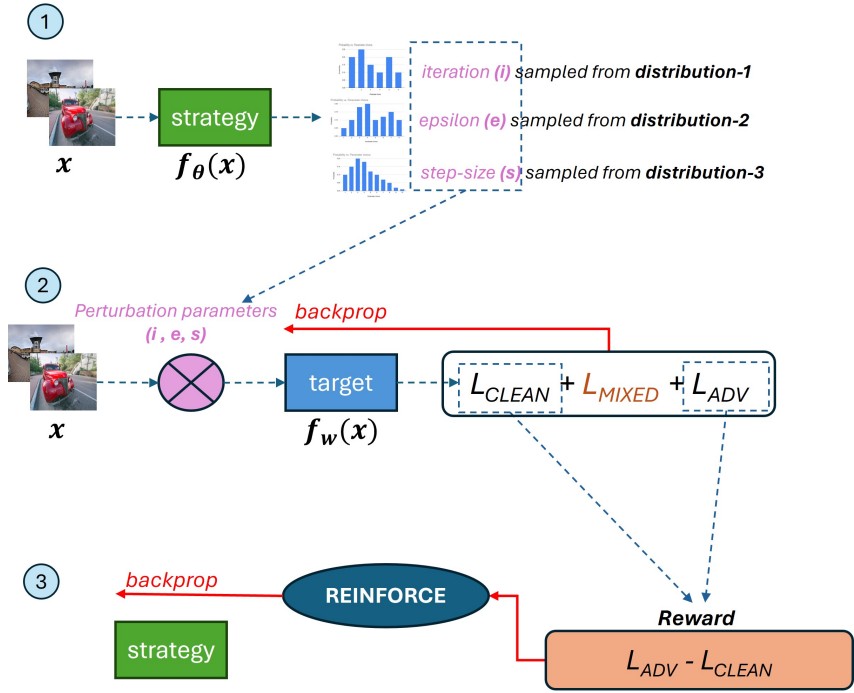

Figure 8: Strategy and Target Network interaction during ASTrA pretraining phase.

Figure 8 illustrates the ASTrA framework involving an adaptive attack strategy network interacting with a target network in an adversarial training setup. The framework focuses on dynamically crafting adversarial examples and aligning the clean and adversarial representations for robust learning, explained in the following steps.

1. The strategy network generates probability distributions for each perturbation parameter from which iteration ($i$), epsilon ($e$), and step size ($s$) are sampled. The perturbation (attack) parameters are sampled according to a *conditional probability distribution*, given by $p(a|x, \theta)$.

2. These perturbed parameters ($i$, $e$, $s$) are used to craft adversarial examples from the input images for target network. The target network feed-forward both clean and adversarially perturbed images and computes three loss terms - **(a)** contrastive loss on clean image views ($\text{Loss}_{\text{NT\_CLEAN}}$), **(b)** contrastive loss on perturbed views ($\text{Loss}_{\text{NT\_ADV}}$), and **(c)** contrastive loss on clean to perturbed views ($\text{Loss}_{\text{NT\_MIXED}}$) and back-propagated to target network.

3. The clean loss **(a)** and adversarial loss **(b)** terms are used to compute reward for strategy network. This computed reward is fed to REINFORCE (Williams (1992)) algorithm to compute gradients required to update the strategy network. The sampling process $a \sim p(a|x; \theta)$ is not differentiable with respect to $\theta$, making traditional gradient-based optimization methods inapplicable. The REINFORCE algorithm is specifically designed to handle such situations by using the **log-derivative trick**, enabling gradient-based optimization even when sampling is involved. The objective of the strategy network is to maximize the expected reward $J(\theta)$, which is defined as:

$$J(\theta) = \mathbb{E}_{x \sim D}\left[\mathbb{E}_{a \sim p(a|x;\theta)}\left[R_{\text{strategy}}(x, a; \theta)\right]\right].\tag{8}$$

To compute the gradient of $J(\theta)$ with respect to $\theta$, we use the property of probability distributions:

$$\nabla_\theta p(a|x; \theta) = p(a|x; \theta)\nabla_\theta \log p(a|x; \theta).\tag{9}$$

The gradient of $J(\theta)$ is expanded as:

$$\nabla_\theta J(\theta) = \mathbb{E}_{x \sim D} \left[ \nabla_\theta \mathbb{E}_{a \sim p(a|x;\theta)} \left[ R_{\text{strategy}}(x, a; \theta) \right] \right]. \tag{10}$$

Using the log-derivative trick, the inner gradient is expressed as:

$$\nabla_\theta \mathbb{E}_{a \sim p(a|x;\theta)} \left[ R_{\text{strategy}} \right] = \mathbb{E}_{a \sim p(a|x;\theta)} \left[ R_{\text{strategy}} \nabla_\theta \log p(a|x;\theta) \right]. \tag{11}$$

Substituting back, the gradient of $J(\theta)$ becomes:

$$\nabla_\theta J(\theta) = \mathbb{E}_{x \sim D} \left[ \mathbb{E}_{a \sim p(a|x;\theta)} \left[ R_{\text{strategy}}(x, a; \theta) \nabla_\theta \log p(a|x;\theta) \right] \right]. \tag{12}$$

The reward $R_{\text{strategy}}$ scales the gradient update, encouraging the strategy network to favor attack parameters $a$ that yield higher rewards. This ensures that the strategy network learns attack strategies that balance adversarial robustness and generalization.

### A.1.1 EFFECT OF STRATEGY NETWORK ON TRAINING STABILITY

The incorporation of the strategy network makes the training process adaptive and stable by dynamically adjusting attack parameters at the instance level, leveraging the learning dynamics of the target network at each step through observations of standard and adversarial loss terms. This stability is evident based on the smoothness of the loss landscape visualization in Figure 11. Additionally, attack parameter bin configurations identified empirically for CIFAR-10 are successfully reused for STL-10 and ImageNet100, demonstrating that the model is not sensitive to initial bin configurations and consistently adapts toward convergence.

### A.1.2 ROLE OF $L_{\text{CLEAN}}$ AND $L_{\text{ADV}}$

The reward function $R_{\text{strategy}}$ incorporates both $L_{\text{adv}}$ and $L_{\text{clean}}$:

$$R_{\text{strategy}}(x, a; \theta) = \alpha L_{\text{adv}}(x, a; \theta) - \gamma L_{\text{clean}}(x, w_{\text{fixed}}). \tag{13}$$

Here:

- $L_{\text{adv}}$ depends on $\theta$, as adversarial examples $x_{\text{adv}}$ are crafted using attack parameters $a$, which are influenced by $\theta$.
- $L_{\text{clean}}$ does **not** depend on $\theta$, as it is computed using clean data and the fixed target network $w_{\text{fixed}}$.

Substituting $R_{\text{strategy}}$ into the gradient:

$$\begin{aligned}
\nabla_\theta J(\theta) &= \mathbb{E}_{x \sim D} \left[ \mathbb{E}_{a \sim p(a|x;\theta)} \left[ \alpha L_{\text{adv}}(x, a; \theta) \nabla_\theta \log p(a|x;\theta) \right] \right] \\
&\quad - \mathbb{E}_{x \sim D} \left[ \mathbb{E}_{a \sim p(a|x;\theta)} \left[ \gamma L_{\text{clean}}(x, w_{\text{fixed}}) \nabla_\theta \log p(a|x;\theta) \right] \right].
\end{aligned} \tag{14}$$

### A.1.3 EFFECT OF $L_{\text{CLEAN}}$

While $L_{\text{clean}}$ does not directly depend on $\theta$, it indirectly affects the updates to $\theta$ by scaling the reward $R_{\text{strategy}}$:

- Large $L_{\text{clean}}$ reduces $R_{\text{strategy}}$, discouraging attack strategies that degrade clean performance.
- Small $L_{\text{clean}}$ increases $R_{\text{strategy}}$, reinforcing attack strategies that preserve generalization.

This ensures that the strategy network learns attack parameters $a$ that balance robustness (via $\alpha L_{\text{adv}}$) and generalization (via $\gamma L_{\text{clean}}$).

### A.1.4 FINAL GRADIENT EXPRESSION

The final gradient is:

$$\begin{aligned}
\nabla_\theta J(\theta) &= \mathbb{E}_{x \sim D} \left[ \mathbb{E}_{a \sim p(a|x;\theta)} \left[ \alpha L_{\text{adv}}(x, a; \theta) \nabla_\theta \log p(a|x;\theta) \right] \right] \\
&\quad - \mathbb{E}_{x \sim D} \left[ \mathbb{E}_{a \sim p(a|x;\theta)} \left[ \gamma L_{\text{clean}}(x, w_{\text{fixed}}) \nabla_\theta \log p(a|x;\theta) \right] \right].
\end{aligned} \tag{15}$$

This ensures the updates to $\theta$ produce controlled attack strategies that improve adversarial robustness while maintaining clean accuracy.

A.2   ASTRA ALGORITHM

Below are the details of the proposed ASTrA algorithm:

---

**Algorithm 1** ASTrA Algorithm

---

**Require:** Training dataset $\mathcal{D}$, target model $f_w$, strategy model $s_\theta$
  1: **for** each epoch **do**
  2:     **for** each batch $B$ in $\mathcal{D}$ **do**
  3:         $x \leftarrow$ augmented views of images in $B$
  4:         **if** update interval reached **then**
  5:             Set $f_w$ to eval mode, $s_\theta$ to train mode
  6:             $a \leftarrow s_\theta(x)$                                      ▷ Sample attack parameters
  7:             $x_{adv} \leftarrow \text{PGD}(f_w, x, a)$                     ▷ Generate adversarial examples
  8:             $r \leftarrow \text{ComputeReward}(f_w, x, x_{adv})$
  9:             REINFORCEUPDATE($s_\theta, a, r, x$)
 10:         **end if**
 11:         Set $f_w$ to train mode, $s_\theta$ to eval mode
 12:         $a \leftarrow s_\theta(x)$                                        ▷ Select attack parameters
 13:         $x_{adv} \leftarrow \text{PGD}(f_w, x, a)$                         ▷ Generate adversarial examples
 14:         $z \leftarrow f_w(x, \text{'normal'})$                            ▷ Clean features
 15:         $z_{adv} \leftarrow f_w(x_{adv}, \text{'pgd'})$                   ▷ Adversarial features
 16:         $\mathcal{L}_{clean} \leftarrow \text{NT-Xent}(z)$               ▷ Contrastive loss on clean samples
 17:         $\mathcal{L}_{adv} \leftarrow \text{NT-Xent}(z_{adv})$           ▷ Contrastive loss on adversarial samples
 18:         $\mathcal{L}_{sim} \leftarrow \text{NT-Xent}([z, z_{adv}])$      ▷ Similarity loss
 19:         $\mathcal{L} \leftarrow (\mathcal{L}_{clean} + \mathcal{L}_{adv})/2 + \lambda\mathcal{L}_{sim}$
 20:         Compute gradients of $\mathcal{L}$ with respect to $f_w$
 21:         Update $f_w$ parameters using computed gradients
 22:     **end for**
 23: **end for**

---

**Algorithm 2** ComputeReward Function

---

  1: **function** COMPUTEREWARD($f_w, x, x_{adv}$)
  2:     $z \leftarrow f_w(x, \text{'normal'})$                              ▷ Clean features
  3:     $z_{adv} \leftarrow f_w(x_{adv}, \text{'pgd'})$                     ▷ Adversarial features
  4:     $\mathcal{L}_{clean} \leftarrow \text{NT-Xent}(z)$
  5:     $\mathcal{L}_{adv} \leftarrow \text{NT-Xent}(z_{adv})$
  6:     $\mathcal{L}_{sim} \leftarrow \text{NT-Xent}([z, z_{adv}])$
  7:     $r \leftarrow w_1\mathcal{L}_{adv} - w_2\mathcal{L}_{clean}$
  8:     **return** $r$
  9: **end function**

---

**Algorithm 3** REINFORCE Update

---

  1: **function** REINFORCEUPDATE($s_\theta, a, r, x$)
  2:     Compute log probability: $\log \pi \leftarrow \log s_\theta(a|x)$
  3:     Compute gradient: $\nabla_\theta J(\theta) \leftarrow r\nabla_\theta \log \pi$
  4:     Update parameters: $\theta \leftarrow \theta + \alpha\nabla_\theta J(\theta)$
  5: **end function**

---

---

**Algorithm 4** PGD Attack

---

1: Initialize $\delta \sim \text{Uniform}(-\epsilon, \epsilon)$
2: **for** $k = 1$ to $K$ **do**
3:     Compute gradient: $g \leftarrow \nabla_\delta \mathcal{L}_{\text{adv}}(f_w(x + \delta))$
4:     Update perturbation: $\delta \leftarrow \delta + \alpha \cdot \text{sign}(g)$
5:     Project perturbation: $\delta \leftarrow \text{clip}(\delta, -\epsilon, \epsilon)$
6:     Ensure valid pixel range: $x^{adv} \leftarrow \text{clip}(x + \delta, 0, 1)$
7: **end for**

---

### A.3   CLEAN AND ADVERSARIAL DISTRIBUTION SAMPLE ALIGNMENT - ROCL VS ASTRA

As shown in Fig. 9, RoCL and ASTrA differ in how they align clean and perturbed distributions using contrastive loss. RoCL creates an augmented view from the original image, then perturbs it to generate an adversarial view. The contrastive loss is applied between the original image, clean augmented view, and adversarial view, resulting in a less structured alignment with only one adversarial view involved. In contrast, ASTrA generates two augmented views, each paired with its

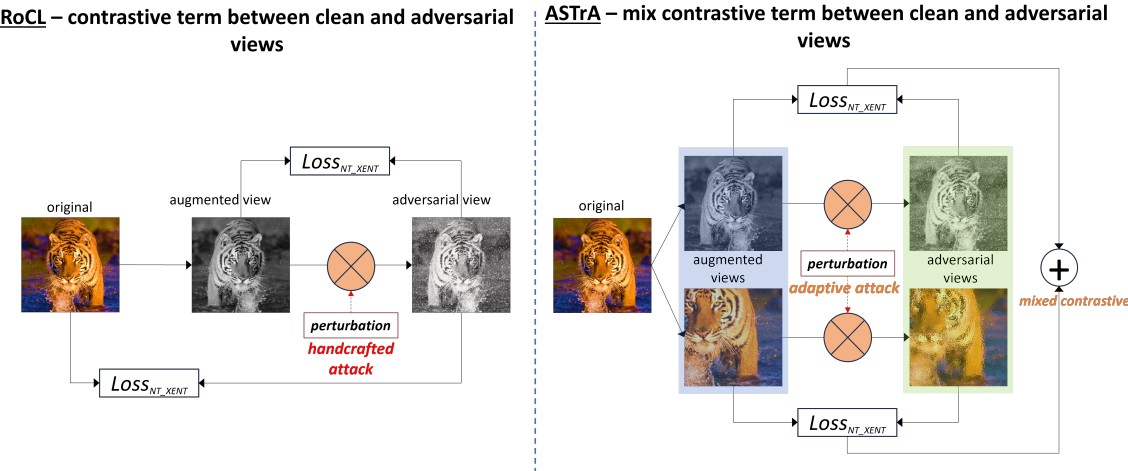

Figure 9: clean and adversarial distribution alignment approach is compared between RoCL Kim et al. (2020) and ASTrA. *For clarity, only clean to adversarial contrastive loss is shown.*

corresponding adversarial view. The contrastive loss is computed between each clean view and its adversarial counterpart, enforcing a more direct and balanced alignment. This "mixed contrastive" objective in ASTrA enhances the alignment between clean and perturbed samples, improving robustness without compromising generalization to clean data. ASTrA's approach is more effective due to this direct pairing, allowing the model to adapt better to diverse adversarial scenarios and generalize more effectively than RoCL's single perturbation method.

### A.4   EXTENDED ANALYSIS AND ABLATIONS

### A.4.1   COMPUTATION ANALYSIS OF ASTRA

The Table 7 highlights the computation analysis of onboarding different strategy networks within the ASTrA framework. The results show that adding a strategy network introduces a slight increase in computation time. For instance, the compute time increases from 20.5 hours for the ACL baseline (which lacks a strategy network) to 23.4 hours for ResNet18, which is the largest architecture in this analysis. This represents an additional compute overhead of less than 3 hours. For smaller networks such as MobileNetV1 or CustomCNN, the increase in compute time is even smaller, around 1 to 1.5 hours. These results indicate that the computational overhead introduced by the strategy network remains minimal and manageable in all cases.

Table 7: Ablation (SLF evaluation) on Strategy network choices. Compute-robustness trade-off. Training conducted on single H100 GPU. ASTrA tends network agnostic for strategy network and even with smaller architectures it outperform ACL Jiang et al. (2020) and choices of networks like ResNet10, EfficientNet-B0, and DenseNet-121 achieves SoTA.

| Method | Strategy Network/Parameters | AA | RA | SA | Compute Time (hrs.) |
|--------|------------------------------|-------|-------|-------|---------------------|
| ACL | - | 37.62 | 40.02 | 79.32 | 20.50 |
| ASTrA | CustomCNN (5-layers)/2.5M | 45.22 | 53.12 | 80.02 | 21.10 |
| ASTrA | MobileNetV1/4.2M | 45.38 | 53.35 | 80.21 | 21.60 |
| ASTrA | ResNet10/5.1M | 45.80 | 53.63 | 80.32 | 21.75 |
| ASTrA | EfficientNet-B0/5.3M | 45.94 | 53.86 | 80.40 | 21.75 |
| ASTrA | DenseNet-121/7.98M | 46.05 | 53.88 | 80.48 | 22.60 |
| ASTrA | ResNet18/11.7M | **46.40** | **54.02** | **80.54** | 23.40 |

Referring Table 7, ASTrA proves to be network-agnostic, achieving consistent performance improvements across both parametric and non-parametric strategy network choices. Regardless of the architecture, ASTrA outperforms the ACL baseline (Jiang et al. (2020)), showcasing its versatility and robustness. Smaller, lightweight architectures such as MobileNetV1 and CustomCNN still achieve competitive performance, while larger architectures such as ResNet10, EfficientNet-B0, and DenseNet-121 improve state-of-the-art results.

The results also reveal a positive trend where increasing the complexity of the strategy network leads to incremental gains in adversarial and robust accuracy metrics. This demonstrates ASTrA's ability to leverage the capacity of various strategy networks effectively, reinforcing its robustness and generalization capabilities while keeping computational overhead minimal.

### A.4.2 COMPARISON OF ASTRA++ WITH METHODS HAVING POST-PROCESSING VERSIONS

The DYNACL++ (Luo et al. (2023)) and DYNACL-AIR++ (Xu et al. (2024)) methods extend two-stage self-supervised adversarial training by introducing a third post-processing stage to enhance representation robustness. This additional stage involves generating pseudo-labels using clustering on pretraining embeddings, followed by Linear Probing and Adversarial Full Finetuning (LP-AFF Kumar et al.). ASTrA++ which is longer pretraining (2000 epochs) version of ASTrA focuses solely on extended pretraining to improve performance without relying on pseudo-labels or additional training phases. We compare ASTrA++ with DYNACL++ and DYNACL-AIR++ in Table 8. Performance of DYNACL-AIR method is compared with ASTrA and other methods and incorporated in respective tables in updated manuscript.

Table 8 compares ASTrA++ with DYNACL++ and DYNACL-AIR++ on CIFAR-10, CIFAR-100, and STL-10 under SLF and AFF evaluation protocols. Despite being limited to a two-stage training framework, ASTrA++ demonstrates improved performance in AFF metrics across all datasets, highlighting the efficacy of extended pretraining in achieving robust and generalized representations. For SLF evaluation, ASTrA++ achieves performance comparable to the state-of-the-art (SoTA), demonstrating its ability to match or exceed the robustness of methods that incorporate additional post-processing stages.

***ASTrA++ does exhibit some limitations***, particularly in SLF results, where the improvements are less effective compared to its gains in AFF metrics. This suggests that while extended pretraining is effective for adversarial robustness, it may not fully address the requirements for improving standard linear evaluation scenarios. Future investigations could explore integrating post-processing stages, such as pseudo-label-based adversarial finetuning, to further enhance ASTrA++'s performance in SLF settings while retaining its strengths in adversarial robustness.

ASTrA can be extended with post-processing used by DYNACL++ (Luo et al. (2023)) and DYNACL-AIR++ (Xu et al. (2024)) however it may considered improved version of ASTrA which violets the submission policy.

Table 8: SLF and AFF evaluation on CIFAR10, CIFAR100, and STL10. ++ in DYNACL++ and DYNACL-AIR++ indicates additional post-processing phase which uses clustering following by Pseudo Adversarial Training. ASTrA++ denotes longer pre-training for 2000 epochs.

| Dataset | Pre-training | SLF | | AFF | |
|---|---|---|---|---|---|
| | | AA (%) | SA (%) | AA (%) | SA (%) |
| CIFAR-10 | DynACL++ | 46.46 | 79.81 | 50.31 | 81.94 |
| | DynACL-AIR++ | **46.99** | **81.80** | 50.65 | 82.36 |
| | ASTrA++ | 46.92 | 80.46 | **50.84** | **83.72** |
| CIFAR-100 | DynACL++ | 20.07 | 52.26 | 25.21 | 57.30 |
| | DynACL-AIR++ | 20.61 | **53.93** | 25.48 | 57.57 |
| | ASTrA++ | **21.95** | 53.58 | **26.45** | **60.25** |
| STL-10 | DynACL++ | 47.21 | 70.93 | 41.84 | 72.36 |
| | DynACL-AIR++ | 47.90 | 71.44 | 44.09 | 72.42 |
| | ASTrA++ | **48.21** | **78.72** | **50.15** | **79.70** |

### A.4.3 LEARNING RATE OF STRATEGY NETWORK

Table 9 shows the critical role of the learning rate in optimizing ASTrA's strategy network during the pretraining stage, directly influencing the effectiveness of the adaptive attack strategy. The performance across CIFAR10 and STL10 reveals that low learning rates $(0.001, 0.01)$ hinder the strategy network's ability to explore and exploit optimal adversarial attacks, leading to under-performance in both AA and RA.

Table 9: Effect of learning rates for the Strategy network of ASTrA. Adversarial full finetuning is performed.

| Learning Rate | CIFAR10 | | | STL10 | | |
|---|---|---|---|---|---|---|
| | AA | RA | SA | AA | RA | SA |
| 0.001 | 47.65 | 49.86 | 79.50 | 46.30 | 49.80 | 77.62 |
| 0.01 | 49.36 | 52.54 | 81.20 | 48.15 | 51.28 | 79.30 |
| 0.1 | **50.84** | **54.90** | **82.68** | **49.65** | **52.40** | **80.20** |
| 0.5 | 50.20 | 54.10 | 82.10 | 49.10 | 51.95 | 79.85 |

The sharp performance gains at a learning rate of 0.1 indicate that this value strikes the ideal balance between exploration and exploitation within the adversarial space, allowing ASTrA's strategy network to generate more adaptive and potent perturbations. At 0.5, the marginal decline in performance suggests that too high a learning rate disrupts the stability of adaptive attacks, potentially leading to overly aggressive perturbations that compromise alignment with the clean distribution. This analysis highlights the role of learning rate in achieving optimal adaptive attack strategies for ASTrA's towards a balanced optimization of adversarial robustness and generalization.

### A.4.4 PROJECTOR HEAD OF TARGET NETWORK

The projector head ablation results (refer Fig. 10) provide valuable insights into the impact of architectural choices for the target network during ASTrA's pretraining. The comparison between a smaller projection head (512, 512) and a Bottleneck configuration (2048, 512) across CIFAR10, CIFAR100, and STL10 demonstrates that the Bottleneck consistently outperforms the smaller head across AA, RA, and SA metrics.

The performance improvement of the Bottleneck can be attributed to its larger latent dimensionality, which enables richer representation learning during adversarial self-supervised pretraining, leading to enhanced adversarial robustness and generalization on clean data. These results emphasize the importance of projection head capacity in optimizing adversarial and standard accuracy during pretraining.

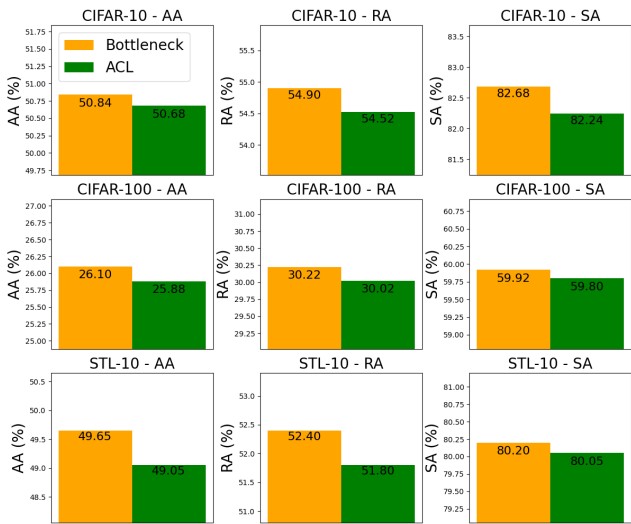

Figure 10: Bottleneck projector head (2014, 512) shows improvement over ACL's project head (512, 512).

### A.4.5 BATCH UPDATES FREQUENCY OF STRATEGY NETWORK

Table 10 illustrates the impact of the frequency of strategy network updates relative to target network updates on performance metrics. The frequency parameter controls how often the strategy network is updated relative to the target network. Specifically, the strategy network is updated every time the target network has been updated a certain number of times, as dictated by the frequency value. From the table, it is clear that a frequency of 10 leads to the best performance across all metrics

Table 10: Effect of batch update frequency for the Strategy network of ASTrA. Adversarial full finetuning is performed.

| Frequency | CIFAR10 | | | STL10 | | |
|---|---|---|---|---|---|---|
| | AA | RA | SA | AA | RA | SA |
| 1 | 47.70 | 50.10 | 79.90 | 46.80 | 48.25 | 78.44 |
| 5 | 50.34 | 54.58 | 82.12 | 49.25 | 52.02 | 79.86 |
| 10 | **50.84** | **54.90** | **82.68** | **49.65** | **52.40** | **80.20** |
| 20 | 50.50 | 54.65 | 82.40 | 49.50 | 52.05 | 80.00 |
| 50 | 50.45 | 54.62 | 82.35 | 49.45 | 52.22 | 80.05 |

(AA, RA, SA), indicating that this update interval strikes the right balance between exploration and exploitation in the strategy network. This frequency allows the strategy model to maintain a balance between exploring new adversarial strategies and stabilizing around effective attacks without reacting too quickly to small changes in the target network. Overly frequent updates may cause the strategy network to overfit to short-term changes in the target network, leading to less effective adversarial attacks overall. The adversarial examples generated by the network are not challenging enough for the target network to generalise better. The frequent changing of the attack strategies, due to frequent strategy network updates, provides less chance for the target network to adapt to the attacks, leading to a drop in performance. Conversely, updating the strategy network too frequently (e.g., at a frequency of 20 or 50) leads to a slight drop in performance, particularly in AA and RA. Although these settings still perform close to the optimal, the slight decrease suggests that less frequent updates may hinder the strategy network's ability to adapt rapidly enough to evolving target network behavior.

A.4.6   DISCRETIZATION OF THE ATTACK PARAMETERS

To find the suitable granularity of attack parameter bins on downstream performance, we earlier conducted experiments by varying the discretization levels of perturbation $\epsilon$, PGD iterations, and step size. The ablations are in Tables 11, 12, and 13 with following analysis.

Table 11: Ablation (SLF evaluation) on discretization of the attack parameter - perturbation.

| Approach | Bins (Perturbation) | AA | RA | SA |
|---|---|---|---|---|
| small-bins | $[3, 7, 11, 15]$ | 38.10 | 41.00 | 79.30 |
| small-bins | $[3, 5, 7, 9, 11, 13, 15]$ | 40.18 | 42.12 | 79.62 |
| original | $[3, 4, 5, 6, 7, 8...., 13, 14, 15]$ | **46.40** | **54.02** | **80.54** |
| large-bins | $[3, 3.5, 4, 4.5..., 14, 14.5, 15]$ | 44.88 | 52.05 | 79.38 |
| large-bins + 2k epochs | $[3, 3.5, 4, 4.5..., 14, 14.5, 15]$ | 46.34 | 54.00 | 80.36 |

Table 12: Ablation (SLF evaluation) on discretization of the attack parameter - step size.

| Approach | Bins (Step-size) | AA | RA | SA |
|---|---|---|---|---|
| small-bins | $[1, 3, 5]$ | 37.90 | 40.22 | 79.55 |
| original | $[1, 2, 3, 4, 5, 6]$ | **46.40** | **54.02** | **80.54** |
| large-bins | $[1, 1.5, 2, 2.5, .., 5.5, 6]$ | 45.02 | 53.06 | 79.12 |
| large-bins + 2k epochs | $[1, 1.5, 2, 2.5, .., 5.5, 6]$ | 46.37 | 53.92 | 80.20 |

**Effect of Coarser Discretization (Smaller Bins)** Using coarser bins for attack parameters simplifies the action space for the strategy network but limits its ability to find the most effective attack strengths. As shown in the tables, when we use smaller bins (e.g., $[3, 7, 11, 15]$ for perturbation $\epsilon$, refer Table 11), there is a noticeable decrease in adversarial accuracy (AA) and robust accuracy (RA). Specifically, AA drops from 46.40% (original bins) to 38.10% with coarser bins for $\epsilon$, though better than baseline ACL Jiang et al. (2020). This confirms that a limited set of attack parameter choices hampers the strategy network's capacity to adaptively challenge the model, leading to sub-optimal robustness.

Table 13: Ablation (SLF evaluation) on discretization of the attack parameter - PGD iterations.

| Approach | Bins (PGD iterations) | AA | RA | SA |
|---|---|---|---|---|
| small-bins | $[3, 7, 11, 14]$ | 38.20 | 40.80 | 79.22 |
| small-bins | $[3, 5, 7, 9, 11, 13]$ | 39.80 | 41.10 | 79.80 |
| original | $[3, 4, 5, 6, 7, 8...., 13, 14]$ | **46.40** | **54.02** | **80.54** |

**Effect of Finer Discretization (Larger Bins)** Introducing finer bins increases the granularity of attack parameter choices, potentially allowing the strategy network to find more optimal strategies. However, as observed, the performance gains with larger bins are marginal compared to the original settings. For instance, with finer bins for step size (Table 12), AA improves slightly to 45.02%, but does not surpass the original setting. Moreover, the computational complexity increases due to the expanded action space, which may require longer training to converge. Notably, when we extend the pretraining to 2000 epochs with larger bins, the model attains results comparable to the original settings (e.g., AA of 46.34% vs. 46.40% for perturbation in Table 11), indicating that longer training can compensate for the increased complexity.

**Empirical Findings and Transferability** Through these experiments, we found that the original bin settings offer a good balance between performance and efficiency. The optimal bin settings for all three attack parameters were determined empirically on CIFAR10 and successfully applied to other datasets without significant performance degradation. This suggests that the optimal parameters are transferable and not highly sensitive to dataset-specific characteristics, enhancing the practicality of our method across different domains.

**Limitations on Bin Approach** While the empirically found parameters demonstrate transferability, ASTrA currently lacks a proven foundation for selecting optimal bin ranges. Dynamically adapting the bins for attack parameters during training based on learning dynamics of target models is one

of the possibility. Incorporating adaptive binning strategies for parameter selection could further improve performance and efficiency. As a plug-and-play framework, ASTrA can be extended in future work to include these capabilities, potentially enhancing its adaptability and robustness.

### A.4.7 Explanation on exploration-exploitation visualization

The transition from exploration to exploitation in ASTrA is caused by the Strategy Network's adaptive learning process as it updates its policy to maximize expected rewards using the REINFORCE algorithm. There is no explicit endpoint for exploration; instead, as the network learns which attack strategies are most effective, it gradually increases the probability of selecting those strategies, naturally shifting toward exploitation.

As the Strategy Network continuously adapts to gradients from the Target Network, there is no definitive boundary where exploration ends, and exploitation begins. The delineation is purely illustrative, serving to emphasize the conceptual transition. Post this point, the model's selection behavior becomes increasingly exploitative, predominantly favoring specific parameter values optimized for Target Model training. This concept is further visualized in Figure 11, where the parameter values are visualized discretely across training epochs. The figure clearly demonstrates a significant increase in the proportion of samples selecting optimal parameter values as the model progresses, highlighting the transition to the exploitation phase.

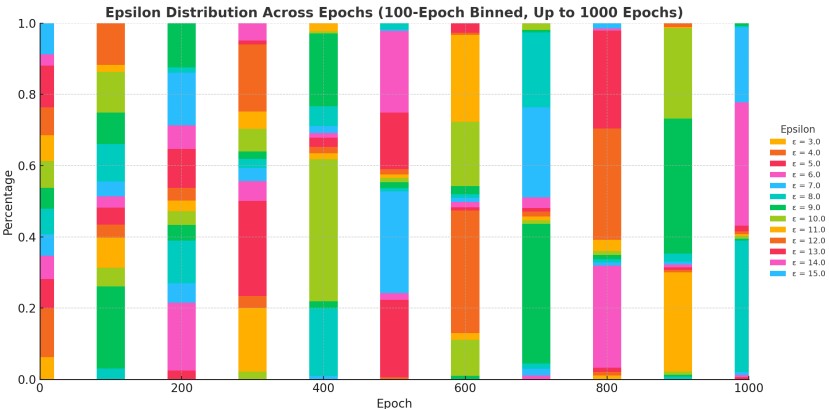

Figure 11: Exploration-Exploitation alternate visualization for Fig 6.

### A.5 Loss Landscape Analysis

To further support the effectiveness of ASTrA's adaptive strategy, the loss landscape comparison, presented in Appendix (Figure 12), demonstrates that ASTrA achieves a flatter loss landscape compared to ACL. This flatter landscape indicates better generalization and robustness, showing how ASTrA not only improves adversarial accuracy but also maintains high performance on clean data. This adaptability and dynamic optimization make ASTrA a more effective solution than static or randomly exploring strategies, demonstrating its ability to find optimal attack policies without relying on brute-force methods.

### A.6 ASTrA and Adversarial Curriculum Learning

ASTrA shares a fundamental similarity with adversarial curriculum learning (A-CL) approaches such as SAT (Sitawarin et al. (2021)) and CAT (Cai et al. (2018)), in that it seeks to improve adversarial training by dynamically adjusting the strength of adversarial examples during training. Both ASTrA and A-CL approaches aim to enhance model robustness while maintaining generalization by optimizing the process of adversarial example generation over time. The motivation behind ASTrA partially overlaps with A-CL. While A-CL methods focus on incrementally increasing attack difficulty (weaker attacks to stronger attacks) to prevent catastrophic forgetting and achieve smooth transitions, ASTrA is motivated by creating an autonomous and adaptive attack framework to establish instance level attack parameters based on learning dynamics of network itself.

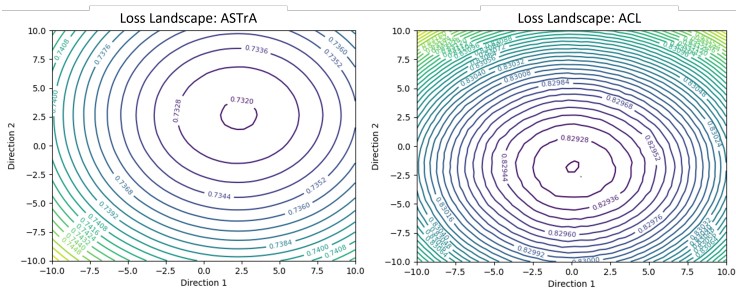

Figure 12: ASTrA vs ACL Loss landscape comparison.

**How ASTrA differs from adversarial curriculum approaches?** - ASTrA's reliance on adaptive, autonomous parameter optimization and its self-supervised foundation mark a significant departure from curriculum-based methods. Following are some important observations.

- **Attack mechanism**: A-CL approaches rely on a predefined or gradually increasing curriculum for adversarial attack strength, which is often heuristic-based whereas ASTrA employs a strategy network guided by reinforcement learning to autonomously adjust attack parameters (e.g., iteration, epsilon, step size). This eliminates the need for predefined rules or heuristics, making ASTrA more adaptable to diverse datasets and training dynamics.

- **Optimization**: ASTrA introduces a reward baed optimization that evaluates the balance between adversarial loss and clean loss, enabling the strategy network to align clean and adversarial distributions effectively. A-CL methods do not typically incorporate such explicit reward-based optimization for attack strategies.

- **Learning approach**: ASTrA designed for a self-supervised setting, more specifically self-supervised adversarial training (self-AT), making it suitable for learning robust representations against adversarial attacks utilizing unlabeled data through its contrastive learning framework. Other side, A-CL methods are designed for supervised settings, where label information often guides the curriculum.

### A.7 CODE REPRODUCIBILITY

The source code of ASTrA is made available at `https://prakashchhipa.github.io/projects/ASTrA` and results can be reproduced with PyTorch 2.0 on CUDA 12.x version.

