# OpenReview forum: "ASTrA: Adversarial Self-supervised Training with Adaptive-Attacks"
_ICLR.cc/2025/Conference — ICLR 2025 Poster_

### Official Review · Reviewer_w4mR · 2024-10-28

**Soundness:** 3
**Presentation:** 2
**Contribution:** 3
**Rating:** 5
**Confidence:** 3

**Summary:**

This paper proposes a novel adaptive attack strategy for adversarial contrastive learning that autonomously discovers optimal attack parameters for generating adversarial examples. By leveraging these parameters in adversarial training, the proposed method enhances adversarial robustness while maintaining standard accuracy.

**Strengths:**

1. The proposed method generates effective adversarial examples by dynamically adjusting attack parameters based on the characteristics of the input data and the model's training dynamics. This adaptive approach outperforms existing methods relying on fixed attack strategies.
2. The proposed method serves as a play-and-play module, allowing seamless integration with existing adversarial contrastive learning methods.

**Weaknesses:**

1. The paper has several readability issues. For instance, equation (3) uses $\theta$ as an input for $L_{\text{clean}}$ while equation (4) switches to using $w^{\text{fixed}}$. This inconsistency causes confusion in understanding the proposed method. Additionally, Table 1 references an undefined [19], making it difficult to verify results. Overall, the writing lacks clarity.
2. The paper lacks clear explanations of the strategy network, a key component of the proposed method.
    - There is insufficient detail on how attack parameters are sampled from the conditional distribution $p(a|x;\theta)$. Since the effectiveness of the adaptive attack strategy depends on this process, a more thorough explanation of how the strategy network determines these parameters is crucial.
    - The interaction between the strategy and target networks is unclear, particularly how $\theta$ is optimized during training. More details are needed on how feedback from the target network and the REINFORCE algorithm update $\theta$ for learning optimal attack strategies.
3. The adaptive attack strategy introduces additional computational overhead. However, there is no analysis of its impact on overall training time compared to existing methods.

**Questions:**

1. Could you explain how attack parameters are sampled from the conditional distribution $p(a|x;\theta)$? Also, could you provide a step-by-step explanation of the interaction between the target and strategy networks? Particularly, it is unclear how $L_{\text{clean}}$ in the reward objective updates $\theta$, as it seems that $\theta$ is not involved in that loss.
2. Could you compare the computational complexity of the proposed method with existing methods? Additionally, could you analyze the computational costs of crafting instance-specific optimal attack strategies and training the strategy network?
3. Could you expand the experimental comparisons to include state-of-the-art methods like DynACL++ and DynACL-AIR++ for a more comprehensive evaluation of the proposed method's strengths and limitations?
    - DynACL-AIR++ is proposed by Xu et al. in Enhancing Adversarial Contrastive Learning via Adversarial Invariant Regularization, NeurIPS 2023.

---

> ### Author Response · Authors · 2024-11-18
> **Review response to Reviewer w4mR**
>
> We thank reviewer w4mR for appreciation and insightful reviews.
>
> **[W1] Readability issues**
>
> Thank you for pointing out on reference in Table 1 and other concerns including clean loss. We have corrected the same in manuscript also improved the readability by revisiting the content and providing added explanation of method with easy-to-understand visualization and step-wise description in appendix.
>
> **[W2, Q1] Interaction between the target and strategy networks and sampling from the conditional distribution**
>
> ### **Interaction between Strategy and Target Networks in ASTrA**
>
> Figure 8 (**Please refer Section A.1 in appendix of updated manuscript**) illustrates the ASTrA framework involving an adaptive attack strategy network interacting with a target network in an adversarial training setup. The framework focuses on dynamically crafting adversarial examples and aligning the clean and adversarial representations for robust learning, explained in the following steps.
>
> **Step 1**: The strategy network generates probability distributions for each perturbation parameter from which iteration (\(i\)), epsilon (\(e\)), and step size (\(s\)) are sampled. The perturbation (attack) parameters are sampled according to a \textit{conditional probability distribution}, given by $p(a|x,\theta)$.
>
> **Step 2**: ThThese perturbed parameters (\(i\), \(e\), \(s\)) are used to craft adversarial examples from the input images for the target network. The target network feed-forwards both clean and adversarially perturbed images and computes three loss terms:
>
> - **(a)** Contrastive loss on clean image views (*$L_{\text{clean}}$*)
> - **(b)** Contrastive loss on perturbed views ( *$L_{\text{adv}}$*), and
> - **(c)** Contrastive loss on clean to perturbed views (*$L_{\text{mixed}}$*)
>
> These loss terms are back-propagated to the target network.
>
> **Step 3**: The clean loss **(a)** and adversarial loss **(b)** terms are used to compute the reward for the strategy network. This computed reward is fed to the REINFORCE [1, Williams, 1992] algorithm to compute gradients required to update the strategy network. The sampling process $( a \sim p(a|x; \theta) )$ is not differentiable with respect to $( \theta )$, making traditional gradient-based optimization methods inapplicable. The REINFORCE algorithm is specifically designed to handle such situations by using the **log-derivative trick**, enabling gradient-based optimization even when sampling is involved. The objective of the strategy network is to maximize the expected reward $J(\theta)$, defined as:
>      \begin{equation}
>       J(\theta) = \mathbb{E}{x \sim D} \left[ \mathbb{E}{a \sim p(a|x; \theta)} \left[ R_{\text{strategy}}(x, a; \theta) \right] \right].
>       \end{equation}
>     - To compute the gradient of $J(\theta)$ with respect to $\theta$, we use the property of probability distributions:
>     \begin{equation}
>     \nabla_\theta p(a|x; \theta) = p(a|x; \theta) \nabla_\theta \log p(a|x; \theta).
>     \end{equation}
>     The gradient of $J(\theta)$ is expanded as:
>     \begin{equation}
>     \nabla_\theta J(\theta) = \mathbb{E}{x \sim D} \left[ \nabla\theta \mathbb{E}{a \sim p(a|x; \theta)} \left[ R{\text{strategy}}(x, a; \theta) \right] \right].
>     \end{equation}
> Using the log-derivative trick, the inner gradient is expressed as:
>     \begin{equation}
>     \nabla_\theta \mathbb{E}{a \sim p(a|x; \theta)} \left[ R{\text{strategy}} \right] = \mathbb{E}{a \sim p(a|x; \theta)} \left[ R{\text{strategy}} \nabla_\theta \log p(a|x; \theta) \right].
>     \end{equation}
> Substituting back, the gradient of $J(\theta)$ becomes:
>     \begin{equation}
>     \nabla_\theta J(\theta) = \mathbb{E}{x \sim D} \left[ \mathbb{E}{a \sim p(a|x; \theta)} \left[ R_{\text{strategy}}(x, a; \theta) \nabla_\theta \log p(a|x; \theta) \right] \right].
>     \end{equation}
> The reward $R_{\text{strategy}}$ scales the gradient update, encouraging the strategy network to favor attack parameters $a$ that yield higher rewards. This ensures that the strategy network learns attack strategies that balance adversarial robustness and generalization.
>
> [1] Williams, Ronald J. "Simple statistical gradient-following algorithms for connectionist reinforcement learning." Machine learning 8 (1992): 229-256.

---

> > ### Author Response · Authors · 2024-11-18
> > **Continuation to [W2, Q1] Interaction between the target and strategy networks and sampling from the conditional distribution**
> >
> > ### **Effect of strategy network on training stability**
> > The incorporation of the strategy network makes the training process adaptive and stable by dynamically adjusting attack parameters at the instance level, leveraging the learning dynamics of the target network at each step through observations of standard and adversarial loss terms. This stability is evident based on the smoothness of the loss landscape visualization in Figure 12 (section A.5 in appendix). Additionally, attack parameter bin configurations identified empirically for CIFAR-10 are successfully reused for STL-10 and ImageNet100, demonstrating that the model is not sensitive to initial bin configurations and consistently adapts toward convergence.
> >
> > ### **Role of *$L_{\text{clean}}$* and *$L_{\text{adv}}$***
> >
> > The reward function $R_{\text{strategy}}$ incorporates both $L_{\text{adv}}$ and $L_{\text{clean}}$:
> > $$
> > R_{\text{strategy}}(x, a; \theta) = \alpha L_{\text{adv}}(x, a; \theta) - \gamma L_{\text{clean}}(x, w_{\text{fixed}}).
> > $$
> > Here:
> > - $L_{\text{adv}}$ depends on $\theta$, as adversarial examples $x_{\text{adv}}$ are crafted using attack parameters $a$, which are influenced by $\theta$.
> > - $L_{\text{clean}}$ does *not* depend on $\theta$, as it is computed using clean data and the fixed target network $w_{\text{fixed}}$.
> >
> > Substituting $R_{\text{strategy}}$ into the gradient that is computed using REINFORCE algorithm using the log-derevative trick:
> > $$
> > \begin{aligned}
> >     \nabla_\theta J(\theta) &= \mathbb{E}{x \sim D} \left[ \mathbb{E}{a \sim p(a|x; \theta)} \left[ \alpha L_{\text{adv}}(x, a; \theta) \nabla_\theta \log p(a|x; \theta) \right] \right]
> >     - \mathbb{E}{x \sim D} \left[ \mathbb{E}{a \sim p(a|x; \theta)} \left[ \gamma L_{\text{clean}}(x, w_{\text{fixed}}) \nabla_\theta \log p(a|x; \theta) \right] \right].
> > \end{aligned}
> > $$
> >
> >
> > ### **Effect of *$L_{\text{clean}}$***
> >
> > While $L_{\text{clean}}$ does not directly depend on $\theta$, we can see that it indirectly affects the updates to $\theta$ by scaling the reward $R_{\text{strategy}}$:
> > - Large $L_{\text{clean}}$ reduces $R_{\text{strategy}}$, discouraging attack strategies that degrade clean performance.
> > - Small $L_{\text{clean}}$ increases $R_{\text{strategy}}$, reinforcing attack strategies that preserve generalization.
> >
> > This ensures that the strategy network learns attack parameters $a$ that balance robustness (via $\alpha L_{\text{adv}}$) and generalization (via $\gamma L_{\text{clean}}$).
> >
> > Our method is detailed further in **section A.1 figure 8 in appendix in manuscript** with details including ASTrA’s steps, strategy network’s role, clean loss in rewards, and training stability.

---

> > > ### Author Response · Authors · 2024-11-18
> > >
> > > **[W3, Q2] Computational overhead of strategy network**
> > >
> > > ### **Computation analysis of ASTrA**
> > > The Table below highlights the computation analysis of onboarding different strategy networks within the ASTrA framework. The results show that adding a strategy network introduces a slight increase in computation time. For instance, the compute time increases from 20.5 hours for the ACL baseline (which lacks a strategy network) to 23.4 hours for ResNet18, which is the largest architecture in this analysis. This represents an additional compute overhead of less than 3 hours. For smaller networks such as MobileNetV1 or CustomCNN, the increase in compute time is even smaller, around 1 to 1.5 hours. These results indicate that the computational overhead introduced by the strategy network remains minimal and manageable in all cases.
> > >
> > > **Table:** Ablation (SLF evaluation) on Strategy network choices. Compute-robustness trade-off. Training conducted on single H100 GPU. ASTrA tends to be network-agnostic for the strategy network, and even with smaller architectures, it outperforms ACL. Choices of networks like ResNet10, EfficientNet-B0, and DenseNet-121 achieve SoTA.
> > >
> > > | **Method** | **Strategy Network/Parameters** | **AA** | **RA** | **SA** | **Compute Time (hrs.)** |
> > > |------------|---------------------------------|--------|--------|--------|-------------------------|
> > > | ACL        | -                               | 37.62  | 40.02  | 79.32  | 20.50                  |
> > > | ASTrA      | CustomCNN (5-layers)/2.5M       | 45.22  | 53.12  | 80.02  | 21.10                  |
> > > | ASTrA      | MobileNetV1/4.2M                | 45.38  | 53.35  | 80.21  | 21.60                  |
> > > | ASTrA      | ResNet10/5.1M                   | 45.80  | 53.63  | 80.32  | 21.75                  |
> > > | ASTrA      | EfficientNet-B0/5.3M            | 45.94  | 53.86  | 80.40  | 21.75                  |
> > > | ASTrA      | DenseNet-121/7.98M              | 46.05  | 53.88  | 80.48  | 22.60                  |
> > > | ASTrA      | ResNet18/11.7M                  | **46.40** | **54.02** | **80.54** | 23.40          |
> > >
> > >
> > > Referring Table above, ASTrA proves to be network-agnostic, achieving consistent performance improvements across both parametric and non-parametric strategy network choices. Regardless of the architecture, ASTrA outperforms the ACL baseline, showcasing its versatility and robustness. Smaller, lightweight architectures such as MobileNetV1 and CustomCNN still achieve competitive performance, while larger architectures such as ResNet10, EfficientNet-B0, and DenseNet-121 improve state-of-the-art results.
> > >
> > > The results also reveal a positive trend where increasing the complexity of the strategy network leads to incremental gains in adversarial and robust accuracy metrics. This demonstrates ASTrA’s ability to leverage the capacity of various strategy networks effectively, reinforcing its robustness and generalization capabilities while keeping computational overhead minimal.

---

> > > > ### Author Response · Authors · 2024-11-18
> > > >
> > > > **[Q3] Comparison with DynACL++ and DynACL-AIR++**
> > > >
> > > > The DYNACL++ [1] and DYNACL-AIR++ [2] methods extend two-stage self-supervised adversarial training by introducing a third post-processing stage to enhance representation robustness. This additional stage involves generating pseudo-labels using clustering on pretraining embeddings, followed by Linear Probing and Adversarial Full Finetuning (LP-AFF [3]). ASTrA++ which is longer pretraining (2000 epochs) version of ASTrA focuses solely on extended pretraining to improve performance without relying on pseudo-labels or additional training phases. We compare ASTrA++ with  DYNACL++ and DYNACL-AIR++ in Table 8. Performance of DYNACL-AIR method is compared with ASTrA and other methods and incorporated in respective tables (Table 1, 2, and 3) in updated manuscript.
> > > >
> > > > Table 8 compares ASTrA++ with DYNACL++ and DYNACL-AIR++ on CIFAR-10, CIFAR-100, and STL-10 under SLF and AFF evaluation protocols. Despite being limited to a two-stage training framework, ASTrA++ demonstrates improved performance in AFF metrics across all datasets, highlighting the efficacy of extended pretraining in achieving robust and generalized representations. For SLF evaluation, ASTrA++ achieves performance comparable to the state-of-the-art (SoTA), demonstrating its ability to match or exceed the robustness of methods that incorporate additional post-processing stages.
> > > >
> > > > **Table 8:** SLF and AFF evaluation on CIFAR10, CIFAR100, and STL10. `++` in DYNACL++ and DYNACL-AIR++ indicates an additional post-processing phase that uses clustering followed by Pseudo Adversarial Training. ASTrA++ denotes longer pre-training for 2000 epochs.
> > > >
> > > > | **Dataset**   | **Pre-training** | **SLF AA (%)** | **SLF SA (%)** | **AFF AA (%)** | **AFF SA (%)** |
> > > > |---------------|------------------|----------------|----------------|----------------|----------------|
> > > > | **CIFAR-10**  | DynACL++         | 46.46          | 79.81          | 50.31          | 81.94          |
> > > > |               | DynACL-AIR++     | **46.99**      | **81.80**      | 50.65          | 82.36          |
> > > > |               | ASTrA++          | 46.92          | 80.46          | **50.84**      | **83.72**      |
> > > > | **CIFAR-100** | DynACL++         | 20.07          | 52.26          | 25.21          | 57.30          |
> > > > |               | DynACL-AIR++     | 20.61          | **53.93**      | 25.48          | 57.57          |
> > > > |               | ASTrA++          | **21.95**      | 53.58          | **26.45**      | **60.25**      |
> > > > | **STL-10**    | DynACL++         | 47.21          | 70.93          | 41.84          | 72.36          |
> > > > |               | DynACL-AIR++     | 47.90          | 71.44          | 44.09          | 72.42          |
> > > > |               | ASTrA++          | **48.21**      | **78.72**      | **50.15**      | **79.70**      |
> > > >
> > > >
> > > > **ASTrA++ does exhibit some limitations**, particularly in SLF results, where the improvements are less effective compared to its gains in AFF metrics. This suggests that while extended pretraining is effective for adversarial robustness, it may not fully address the requirements for improving standard linear evaluation scenarios. Future investigations could explore integrating post-processing stages, such as pseudo-label-based adversarial finetuning, to further enhance ASTrA++'s performance in SLF settings while retaining its strengths in adversarial robustness. Detailed are presented in **section A.4.2 in Appendix**.
> > > >
> > > > ASTrA can be extended with post-processing used by DYNACL++ [1] and DYNACL-AIR++ [2] however it may considered improved version of ASTrA which violets the submission policy.
> > > >
> > > >
> > > > [1] Luo, Rundong, Yifei Wang, and Yisen Wang. "Rethinking the Effect of Data Augmentation in Adversarial Contrastive Learning." The Eleventh International Conference on Learning Representations.
> > > >
> > > > [2] Xu et al. in Enhancing Adversarial Contrastive Learning via Adversarial Invariant Regularization, NeurIPS 2023.
> > > >
> > > > [3] Kumar, Ananya, et al. "Fine-Tuning can Distort Pretrained Features and Underperform Out-of-Distribution." International Conference on Learning Representations.

---

> ### Author Response · Authors · 2024-11-25
> **Summary of our response**
>
> We sincerely thank you for your detailed review and insightful feedback on our paper.
>
> **Summary of Responses:**
>
> - We have addressed all the readability concerns, clarified the interaction between the target and strategy networks, and provided additional computational complexity analysis. Specifically:
>   - Corrected references and equations to improve the clarity of the manuscript.
>   - Added detailed explanations and step-by-step interaction between the target and strategy networks, including how attack parameters are sampled from the conditional distribution and the role of reward optimization using REINFORCE.
>   - Provided computational overhead analysis to highlight ASTrA's efficiency compared to existing methods.
>   - Included comparisons with state-of-the-art methods like DynACL++ and DynACL-AIR++ to demonstrate ASTrA’s strengths.
>
> **Satisfaction with Responses:**
>
> - We believe our responses and added experiments effectively address the concerns. The paper has been revised for clarity, and all questions have been answered comprehensively with supporting empirical evidence.
>
> **Request for Score Improvement:**
>
> - Based on our thorough responses and improvements to the manuscript, we kindly request you to consider increasing the score for our submission. We are happy to address any further questions or suggestions.
>
> Thank you for your thoughtful and constructive feedback.

---

> > ### Comment · Reviewer_w4mR · 2024-11-28
> >
> > Thank you for addressing the questions I raised and resolving the weaknesses I pointed out. However, as demonstrated by the SLF experiment results, the robustness of the representation learned by the proposed method is limited. Therefore, I will maintain the current score.

---

> ### Author Response · Authors · 2024-11-28
>
> We would like to thank the reviewer for their thoughtful feedback. We are responding to ensure there is no misunderstanding regarding what ASTrA and ASTrA++ represent and the SLF/AFF evaluation results.
>
> 1. **ASTrA++ Pretraining:**
>    ASTrA++ employs **longer pretraining (2000 epochs)**, whereas DynACL++ and DynACL-AIR++ utilize a **post-processing phase** (clustering followed by Pseudo Adversarial Training). These approaches are fundamentally different, and ASTrA++ results do not involve post-processing.
>
> 2. **Evaluation Results:**
>  **ASTrA**
>     * ASTrA achieved **state-of-the-art performance** across all datasets (CIFAR-10,CIFAR-100,STL-10) in both **SLF and AFF** evaluations when compared to DynACL and DynACL-AIR, highlighting the robustness of our approach.
>
>     **ASTrA++**
>       * ASTrA++ achieves **state-of-the-art performance** on CIFAR-100 and STL-10 datasets in both **SLF and AFF** evaluations, highlighting its robustness. Unlike DynACL++ and DynACL-AIR++, our approach ASTrA++ **does not apply any post-processing** methods and **demonstrates competitive results solely through extended pretraining**. We note that ASTrA++ could also benefit from post-processing; however, such modifications **are not reported** to comply with **ICLR guidelines prohibiting post-submission changes**.
>
> 3. **Robustness:**
>    The results demonstrate that our method achieves strong representation robustness **solely through pretraining**, without relying on post-processing, underscoring the effectiveness of the longer pretraining approach.
>
> We hope this clarification resolves any ambiguity regarding the robustness achieved by our method. Thank you again for your constructive feedback and engagement. We are looking forward for score upgrades and happy to address any further queries or suggestions.

---

### Official Review · Reviewer_veWg · 2024-11-04

**Soundness:** 3
**Presentation:** 3
**Contribution:** 3
**Rating:** 8
**Confidence:** 3

**Summary:**

This work attempts to improve the generation of adversarial examples for use in the adversarial self-supervised training (self-AT). Self-AT is made difficult by the fact that ground truth labels are not known. The first contribution of this paper consists of a Strategy Network, which is a module that can be incorporated during constrastive training to adaptively select parameters for adversarial example generation, conditioning on both the input sample and the contrastive model's current parameters. The Strategy Network is designed to balance adversarial contrastive loss, which encourages a high dissimilarity between perturbed representations, and clean contrastive loss, which ensures that the Target Network maintains high performance on the clean data distribution. For training the Target Network, a new Mixed Contrastive Loss term is introduced, which prevents robust overfitting by penalizing excess distance between clean and adversarial perturbations. While these modifications to contrastive training allow for an improved tradeoff between strong adversarial perturbations during training and high clean accuracy, they do introduce difficulties in training due to the non-differentiable nature of adversarial example generation. To remedy this, the REINFORCE algorithm is used to estimate the gradients of the Strategy Network's objective function. The complete method is evaluated by pretraining the ResNet18 architecture using three different datasets (CIFAR10, CIFAR100, and STL10), and then finetuning using one of three approaches (standard linear finetuning, adversarial linear finetuning, and adversarial full finetuning). Both clean and robust accuracy are computed, and compared against several other robust contrastive learning methods. ASTrA is shown to improve both robust and clean accuracy over the other training methods tested. An ablation study shows that the individual components each contribute to this improvement, and it is shown how ASTrA can be used as a plug and play component to improve the performance of the other methods.

**Strengths:**

- Using reinforcement learning to optimize adversarial attack generation is well motivated in this paper and seems like a powerful technique to improve robust training. I could imagine this technique being beneficial in applications beyond self-supervised contrastive training.
- The experimental section appears well designed. Several state of the art methods are used as comparison, and four datasets are utilized (CIFAR-10, CIFAR-100, STL-10, and Imagenet-100).
- The plug and play framework presented here could potentially offer a significant leap in robustness for self-supervised contrastive models.
- Code is provided for reproduction purposes.

**Weaknesses:**

- The evaluation section could include more information on the impact of ASTrA on other metrics (like training time or training stability). I would assume that the additional complexity introduced by the strategy network might make the training process more difficult, although I would be interested in hearing if this wasn't the case. I do believe the effects on the training process warrant more explanation in the paper.
- I think one underexplored aspect of this approach is how the discretization of the attack parameters affects the downstream performance of the model. I imagine there would be some sort of tradeoff between training efficiency and performance as you vary the number of parameter choices that the strategy network can make. Were there any experiments where the discretization of the attack strategy was changed?

**Questions:**

- Would the methods in this paper (particularly the strategy network) be suitable for standard adversarial training in a supervised, non-contrastive setting? It seems that adaptively choosing attack parameters might be beneficial in adversarial training more generally.
- I'm not sure I understand what causes the exploration phase to end and the exploitation phase to begin. For example, in Figure 6, why are the dotted lines drawn where they are? The changes don't always seem consistent between the plots. I also found this plot somewhat difficult to understand in general, it wasn't immediately clear to me that the different colors represented different choices of attack parameters.
- Why is Figure 8 a line graph? It seems like it could be more effectively presented as a table.
- How do you think your work relates to the body of research studying adversarial curriculum learning (i.e. [1,2])? At a high level, these works seek to improve adversarial training by adjusting the strength of adversarial examples used during the training process. Do you see ASTrA as having similar motivation and design inspiration to these techniques? How does ASTrA differ from these kinds of curriculum approaches?
- What limitations does your approach have?

[1] Sitawarin, Chawin, Supriyo Chakraborty, and David Wagner. "Sat: Improving adversarial training via curriculum-based loss smoothing." In Proceedings of the 14th ACM Workshop on Artificial Intelligence and Security, pp. 25-36. 2021.

[2] Cai, Qi-Zhi, Min Du, Chang Liu, and Dawn Song. "Curriculum adversarial training." arXiv preprint arXiv:1805.04807 (2018).

---

> ### Author Response · Authors · 2024-11-18
> **Review response to Reviewer veWg**
>
> We thank reviewer veWg for appreciation and insightful reviews.
>
> **[W1] Training complexity of ASTrA**
>
> ### **Effect of strategy network on training stability**
> The incorporation of the strategy network makes the training process adaptive and stable by dynamically adjusting attack parameters at the instance level, leveraging the learning dynamics of the target network at each step through observations of standard and adversarial loss terms. This stability is evident based on the smoothness of the loss landscape visualization in Figure 12 (section A.5 in appendix). Additionally, attack parameter bin configurations identified empirically for CIFAR-10 are successfully reused for STL-10 and ImageNet100, demonstrating that the model is not sensitive to initial bin configurations and consistently adapts toward convergence.
>
>
> ### **Computation analysis of ASTrA**
> The Table below highlights the computation analysis of onboarding different strategy networks within the ASTrA framework. The results show that adding a strategy network introduces a slight increase in computation time. For instance, the compute time increases from 20.5 hours for the ACL baseline (which lacks a strategy network) to 23.4 hours for ResNet18, which is the largest architecture in this analysis. This represents an additional compute overhead of less than 3 hours. For smaller networks such as MobileNetV1 or CustomCNN, the increase in compute time is even smaller, around 1 to 1.5 hours. These results indicate that the computational overhead introduced by the strategy network remains minimal and manageable in all cases.
>
> **Table:** Ablation (SLF evaluation) on Strategy network choices. Compute-robustness trade-off. Training conducted on single H100 GPU. ASTrA tends to be network-agnostic for the strategy network, and even with smaller architectures, it outperforms ACL. Choices of networks like ResNet10, EfficientNet-B0, and DenseNet-121 achieve SoTA.
>
> | **Method** | **Strategy Network/Parameters** | **AA** | **RA** | **SA** | **Compute Time (hrs.)** |
> |------------|---------------------------------|--------|--------|--------|-------------------------|
> | ACL        | -                               | 37.62  | 40.02  | 79.32  | 20.50                  |
> | ASTrA      | CustomCNN (5-layers)/2.5M       | 45.22  | 53.12  | 80.02  | 21.10                  |
> | ASTrA      | MobileNetV1/4.2M                | 45.38  | 53.35  | 80.21  | 21.60                  |
> | ASTrA      | ResNet10/5.1M                   | 45.80  | 53.63  | 80.32  | 21.75                  |
> | ASTrA      | EfficientNet-B0/5.3M            | 45.94  | 53.86  | 80.40  | 21.75                  |
> | ASTrA      | DenseNet-121/7.98M              | 46.05  | 53.88  | 80.48  | 22.60                  |
> | ASTrA      | ResNet18/11.7M                  | **46.40** | **54.02** | **80.54** | 23.40          |
>
>
> Referring Table above, ASTrA proves to be network-agnostic, achieving consistent performance improvements across both parametric and non-parametric strategy network choices. Regardless of the architecture, ASTrA outperforms the ACL baseline, showcasing its versatility and robustness. Smaller, lightweight architectures such as MobileNetV1 and CustomCNN still achieve competitive performance, while larger architectures such as ResNet10, EfficientNet-B0, and DenseNet-121 improve state-of-the-art results.
>
> The results also reveal a positive trend where increasing the complexity of the strategy network leads to incremental gains in adversarial and robust accuracy metrics. This demonstrates ASTrA’s ability to leverage the capacity of various strategy networks effectively, reinforcing its robustness and generalization capabilities while keeping computational overhead minimal.

---

> > ### Author Response · Authors · 2024-11-18
> >
> > **[W2] Discretization of the attack parameters**
> >
> > To find the suitable granularity of attack parameter bins on downstream performance, we earlier conducted experiments by varying the discretization levels of perturbation $\epsilon$, PGD iterations, and step size. The ablations are in Tables 11, 12, and 13.
> >
> > **Table 11:** SLF evaluation on discretization of perturbation  $\epsilon$.
> >
> > | **Approach**          | **Bins (Perturbation)**                          | **AA**   | **RA**   | **SA**   |
> > |------------------------|-------------------------------------------------|----------|----------|----------|
> > | small-bins            | `[3,7,11,15]`                                   | 38.10    | 41.00    | 79.30    |
> > | small-bins            | `[3,5,7,9,11,13,15]`                            | 40.18    | 42.12    | 79.62    |
> > | original              | `[3,4,5,6,7,8,...,13,14,15]`                    | **46.40** | **54.02** | **80.54** |
> > | large-bins            | `[3,3.5,4,4.5,...,14,14.5,15]`                  | 44.88    | 52.05    | 79.38    |
> > | large-bins + 2k epochs | `[3,3.5,4,4.5,...,14,14.5,15]`                  | 46.34    | 54.00    | 80.36    |
> >
> > **Table 12:** SLF evaluation on discretization of step size.
> >
> > | **Approach**          | **Bins (Step-size)**           | **AA**   | **RA**   | **SA**   |
> > |------------------------|-------------------------------|----------|----------|----------|
> > | small-bins            | `[1,3,5]`                     | 37.90    | 40.22    | 79.55    |
> > | original              | `[1,2,3,4,5,6]`               | **46.40** | **54.02** | **80.54** |
> > | large-bins            | `[1,1.5,2,2.5,...,5.5,6]`     | 45.02    | 53.06    | 79.12    |
> > | large-bins + 2k epochs | `[1,1.5,2,2.5,...,5.5,6]`     | 46.37    | 53.92    | 80.20    |
> >
> > **Table 13:** SLF evaluation on discretization of PGD iterations.
> >
> > | **Approach**  | **Bins (PGD iterations)**          | **AA**   | **RA**   | **SA**   |
> > |---------------|------------------------------------|----------|----------|----------|
> > | small-bins    | `[3,7,11,14]`                     | 38.20    | 40.80    | 79.22    |
> > | small-bins    | `[3,5,7,9,11,13]`                 | 39.80    | 41.10    | 79.80    |
> > | original      | `[3,4,5,6,7,8,...,13,14]`         | **46.40** | **54.02** | **80.54** |
> >
> >
> > **Effect of Coarser Discretization (Smaller Bins)** Using coarser bins for attack parameters simplifies the action space for the strategy network but limits its ability to find the most effective attack strengths. As shown in the tables, when we use smaller bins (e.g., $[3,7,11,15]$ for perturbation $\epsilon$, refer Table 11), there is a noticeable decrease in adversarial accuracy (AA) and robust accuracy (RA). Specifically, AA drops from 46.40\% (original bins) to 38.10\% with coarser bins for $\epsilon$, though better than baseline ACL. This confirms that a limited set of attack parameter choices hampers the strategy network's capacity to adaptively challenge the model, leading to sub-optimal robustness.
> >
> > **Effect of Finer Discretization (Larger Bins)**  Introducing finer bins increases the granularity of attack parameter choices, potentially allowing the strategy network to find more optimal strategies. However, as observed, the performance gains with larger bins are marginal compared to the original settings. For instance, with finer bins for step size (Table 12), AA improves slightly to 45.02%, but does not surpass the original setting. Moreover, the computational complexity increases due to the expanded action space, which may require longer training to converge. Notably, when we extend the pretraining to 2000 epochs with larger bins, the model attains results comparable to the original settings (e.g., AA of 46.34% vs. 46.40% for perturbation in Table 13), indicating that longer training can compensate for the increased complexity.
> >
> > **Empirical Findings and Transferability**  Through these experiments, we found that the original bin settings offer a good balance between performance and efficiency. The optimal bin settings for all three attack parameters were determined empirically on CIFAR10 and successfully applied to other datasets without significant performance degradation. This suggests that the optimal parameters are transferable and not highly sensitive to dataset-specific characteristics, enhancing the practicality of our method across different domains.
> >
> > **Limitations on Bin Approach** While the empirically found parameters demonstrate transferability, ASTrA currently lacks a proven foundation for selecting optimal bin ranges. Dynamically adapting the bins for attack parameters during training based on learning dynamics of target models is one of the possibility. Incorporating adaptive binning strategies for parameter selection could further improve performance and efficiency. As a plug-and-play framework, ASTrA can be extended in future work to include these capabilities, potentially enhancing its adaptability and robustness.

---

> > > ### Author Response · Authors · 2024-11-18
> > >
> > > **[Q1] Strategy network in supervised and non-contrastive setting**
> > >
> > > It is feasible to adapt ASTrA's adaptive attack strategy to a supervised, non-contrastive setting by modifying the reward mechanism to incorporate supervised loss terms. This would allow the strategy network to dynamically optimize attack parameters while leveraging label information for improved robustness.
> > >
> > > **[Q2] Exploration to exploitation phase in ASTrA**
> > >
> > > The transition from exploration to exploitation in ASTrA is caused by the Strategy Network's adaptive learning process as it updates its policy to maximize expected rewards using the REINFORCE algorithm. There is no explicit endpoint for exploration; instead, as the network learns which attack strategies are most effective, it gradually increases the probability of selecting those strategies, naturally shifting toward exploitation.
> > >
> > > As the Strategy Network continuously adapts to gradients from the Target Network, there is no definitive boundary where exploration ends, and exploitation begins, as correctly noted by reviewer. The delineation is purely illustrative, serving to emphasize the conceptual transition. Post this point, the model's selection behavior becomes increasingly exploitative, predominantly favoring specific parameter values optimized for Target Model training. **This concept is further visualized in Figure 11 in section A.4.7 in appendix**, where the parameter values are visualized discretely across training epochs. The figure clearly demonstrates a significant increase in the proportion of samples selecting optimal parameter values as the model progresses, highlighting the transition to the exploitation phase.
> > >
> > > **[Q3]  Figure 8 to a line graph**
> > >
> > > Manuscript is updated by replacing Figure 8 to table to effectively showcase attack strategies comparisons.
> > >
> > > **[Q4] Adversarial curriculum learning and ASTrA**
> > >
> > > ASTrA shares a fundamental similarity with adversarial curriculum learning (A-CL) approaches such as SAT [1] and CAT [2], in that it seeks to improve adversarial training by dynamically adjusting the strength of adversarial examples during training. Both ASTrA and A-CL approaches aim to enhance model robustness while maintaining generalization by optimizing the process of adversarial example generation over time. The motivation behind ASTrA partially overlaps with A-CL. While A-CL methods focus on incrementally increasing attack difficulty (weaker attacks to stronger attacks) to prevent catastrophic forgetting and achieve smooth transitions, ASTrA is motivated by creating an autonomous and adaptive attack framework to establish instance level attack parameters based on learning dynamics of network itself.
> > >
> > > **How ASTrA differs from adversarial curriculum approaches?** - ASTrA’s reliance on adaptive, autonomous parameter optimization and its self-supervised foundation mark a significant departure from curriculum-based methods. Following are some important observations.
> > >
> > > **Attack mechanism**:  A-CL approaches rely on a predefined or gradually increasing curriculum for adversarial attack strength, which is often heuristic-based whereas ASTrA employs a strategy network guided by reinforcement learning to autonomously adjust attack parameters (e.g., iteration, epsilon, step size). This eliminates the need for predefined rules or heuristics, making ASTrA more adaptable to diverse datasets and training dynamics.
> > >
> > > **Optimization**:  ASTrA introduces a reward based optimization that evaluates the balance between adversarial loss and clean loss, enabling the strategy network to align clean and adversarial distributions effectively. A-CL methods do not typically incorporate such explicit reward-based optimization for attack strategies.
> > >
> > > **Learning approach**:  ASTrA designed for a self-supervised setting, more specifically self-supervised adversarial training (self-AT), making it suitable for learning robust representations against adversarial attacks utilizing unlabeled data through its contrastive learning framework. Other side, A-CL methods are designed for supervised settings, where label information often guides the curriculum.
> > >
> > > **This section presented in appendix section A.6**
> > >
> > > [1] Sitawarin, Chawin, Supriyo Chakraborty, and David Wagner. "Sat: Improving adversarial training via curriculum-based loss smoothing." In Proceedings of the 14th ACM Workshop on Artificial Intelligence and Security, pp. 25-36. 2021.
> > >
> > > [2] Cai, Qi-Zhi, Min Du, Chang Liu, and Dawn Song. "Curriculum adversarial training." arXiv preprint arXiv:1805.04807 (2018).
> > >
> > > **[Q5] Limitations of ASTrA**
> > >
> > > ASTrA does not dynamically adapts the bins for attack parameters during training based on learning dynamics of target models, as in [W2] Discretization of the attack parameters. ASTrA  does not use any post-processing methods like DYNACL++ [3] **(Sec. A.4.2)**. This both limitation can be future work.
> > >
> > > [3] Luo  et al. "Rethinking the Effect of Data Augmentation in Adversarial Contrastive Learning." ICLR 2023.

---

> ### Author Response · Authors · 2024-11-25
> **Summary of our response**
>
> We sincerely thank you for your detailed review and insightful feedback on our paper.
>
> **Summary of Responses:**
>
> - We have addressed all questions and provided empirical evidence for ASTrA's effectiveness. Specifically:
>   - Detailed analysis of training complexity and stability, showing that ASTrA introduces minimal computational overhead and remains stable across training setups.
>   - Comprehensive experiments on the discretization of attack parameters, demonstrating the trade-offs between efficiency and performance.
>   - Clarifications on how ASTrA could be adapted for supervised settings and how exploration transitions into exploitation in the training process.
>   - Improved presentation of Figure 8 by replacing it with a table and contextualizing ASTrA's relationship to adversarial curriculum learning.
>
> **Satisfaction with Responses:**
>
> - We believe our responses effectively address the concerns raised. We have added clarifications and empirical insights to enhance the paper’s contributions and its alignment with your suggestions.
>
> **Request for Score Improvement:**
>
> - Based on our thorough responses and updates, we kindly request you to consider increasing the score for our submission. We are available to address any further queries or suggestions.
>
> Thank you for your thoughtful and constructive feedback.

---

> > ### Comment · Reviewer_veWg · 2024-11-26
> >
> > I thank the authors for their thorough response. My questions and concerns have all been addressed, and I will be raising my score.

---

> ### Author Response · Authors · 2024-11-26
>
> We thank reviewer veWg for their efforts for going through our response and increasing score.

---

### Official Review · Reviewer_iFgE · 2024-11-06

**Soundness:** 3
**Presentation:** 3
**Contribution:** 3
**Rating:** 6
**Confidence:** 3

**Summary:**

The paper proposes a new method ASTrA for self-supervised training of adversarially robust features. Specifically, the paper introduces several novel and important components to boost the performance of adversarial self-supervised training: mixed contrastive loss and adaptive strategy network. With these two new component, the paper improves the robust accuracy by 1-2% on commonly used datasets like CIFAR10. Ablation study verifies the necessity on these components. Further experiments demonstrates the scalability and potential for wide applications when combining with other methods.

**Strengths:**

1. The paper studies an important topic, self-supervised adversarial training, and proposes novel components to improve the performance of the self-supervised AT.

2. The empirical experiments are extensive, showing improvement of ASTrA over existing method. The ablation study and analysis are thorough, demonstrating the necessity on each component.

**Weaknesses:**

1. Lacking of empirical experiments on larger scale to really reflect the effectiveness of self-supervised AT. As CIFAR10, CIFAR100 and STL-10 are all supervised datasets with clear label, supervised adversarial training may be enough to achieve good robust accuracy. Using unlabel data may be a good way to further improve the performance. However, the paper lacks of the experiments in this aspect.

**Questions:**

1. The last term of reward (eq (4)) does not contain the optimized parameter $\theta$, why is it necessary to add it to the reward function?

2. As the number of parameters of ResNet 18 is already higher than the number of training example times the number of action space, is it possible to use the non-parametric modeling to replace the strategy network?

---

> ### Author Response · Authors · 2024-11-18
> **Review response to Reviewer iFgE**
>
> We thank reviewer iFgE for appreciation and insightful reviews.
>
> **[W1] - empirical experiments on larger scale**
>
> Despite CIFAR10, CIFAR100, and STL-10 being supervised datasets, ASTrA utilizes them without labels during the pretraining stage, as it is a self-supervised adversarial training method. The primary objective of using these datasets is to enable a fair comparison with existing self-supervised adversarial training methods, as most prior works use these datasets as baselines.
>
> **In Section 4.3**, ASTrA's scalability is evaluated by applying it to the larger ImageNet-100 dataset, which contains 120k images with a resolution of 224x224, compared to CIFAR10's 32x32 resolution and STL-10's 96x96 resolution. ASTrA continues to demonstrate consistent robustness on the ImageNet-100 dataset, further validating its scalability and effectiveness on larger, more complex datasets.
>
> **[Q1] The last term of reward (eq (4)) does not contain the optimized parameter $\theta$, why is it necessary to add it to the reward function?**
>
> ### **Role of *$L_{\text{clean}}$* and *$L_{\text{adv}}$***
>
> The reward function $R_{\text{strategy}}$ incorporates both $L_{\text{adv}}$ and $L_{\text{clean}}$:
> $$
> R_{\text{strategy}}(x, a; \theta) = \alpha L_{\text{adv}}(x, a; \theta) - \gamma L_{\text{clean}}(x, w_{\text{fixed}}).
> $$
> Here:
> - $L_{\text{adv}}$ depends on $\theta$, as adversarial examples $x_{\text{adv}}$ are crafted using attack parameters $a$, which are influenced by $\theta$.
> - $L_{\text{clean}}$ does *not* depend on $\theta$, as it is computed using clean data and the fixed target network $w_{\text{fixed}}$.
>
> Substituting $R_{\text{strategy}}$ into the gradient that is computed using REINFORCE algorithm using the log-derevative trick:
> $$
> \begin{aligned}
>     \nabla_\theta J(\theta) &= \mathbb{E}{x \sim D} \left[ \mathbb{E}{a \sim p(a|x; \theta)} \left[ \alpha L_{\text{adv}}(x, a; \theta) \nabla_\theta \log p(a|x; \theta) \right] \right]
>     - \mathbb{E}{x \sim D} \left[ \mathbb{E}{a \sim p(a|x; \theta)} \left[ \gamma L_{\text{clean}}(x, w_{\text{fixed}}) \nabla_\theta \log p(a|x; \theta) \right] \right].
> \end{aligned}
> $$
>
>
> ### **Effect of *$L_{\text{clean}}$***
>
> While $L_{\text{clean}}$ does not directly depend on $\theta$, we can see that it indirectly affects the updates to $\theta$ by scaling the reward $R_{\text{strategy}}$:
> - Large $L_{\text{clean}}$ reduces $R_{\text{strategy}}$, discouraging attack strategies that degrade clean performance.
> - Small $L_{\text{clean}}$ increases $R_{\text{strategy}}$, reinforcing attack strategies that preserve generalization.
>
> This ensures that the strategy network learns attack parameters $a$ that balance robustness (via $\alpha L_{\text{adv}}$) and generalization (via $\gamma L_{\text{clean}}$).
>
> Our method is detailed further in **section A.1 figure 8 in appendix in manuscript** with details including ASTrA’s steps, strategy network’s role, clean loss in rewards, and training stability.
>
> **[Q2] non-parametric modeling**
>
> In ASTrA, the ability to create adaptive adversarial attacks relies heavily on the REINFORCE algorithm, which requires a differentiable, parameterized policy to compute gradients. This setup allows the Strategy Network to learn and update its attack strategies based on reward signals from the Target Network's performance. Non-parametric models, however, lack fixed parameters that can be adjusted through gradient computations. This makes them incompatible with the gradient-based optimization that REINFORCE depends on. Without the capacity to update parameters via gradients, non-parametric methods can't adapt attack strategies in response to the changing state of the Target Network. This incompatibility disrupts the core mechanism of ASTrA, where dynamic adjustment of attack strategies is crucial for effective adversarial training.
>
> While Gaussian Processes [1] offer a non-parametric approach that can, in theory, provide adaptable function approximations, they are impractical for ASTrA's context due to their computational complexity. Gaussian Processes have a computational cost that scales cubically with the number of data points ($\mathcal{O}(n^3)$), making them unsuitable for large datasets common in adversarial training of image models. Additionally, integrating Gaussian Processes into the REINFORCE framework is challenging because computing gradients in high-dimensional spaces (such as those of image data) is computationally intensive and inefficient. Therefore, despite their theoretical capability to enable adaptation, Gaussian Processes are not feasible for ASTrA given the dataset sizes and the need for scalable modeling.
>
> [1] Williams, C. K., & Rasmussen, C. E. (2006). Gaussian processes for machine learning (Vol. 2, No. 3, p. 4). Cambridge, MA: MIT press.
>
> We have added additional computational analysis for different choices of strategy networks is presented in **section A.4.1 table 7 in appendix of manuscript**.

---

> ### Author Response · Authors · 2024-11-25
> **Summary of our response**
>
> We sincerely thank you for your detailed review and insightful feedback on our paper.
>
> **Summary of Responses:**
>
> - We have addressed all questions raised and provided empirical evidence supporting our method. Specifically:
>   - Empirical experiments on larger-scale datasets have been included to demonstrate ASTrA's scalability and effectiveness. We evaluated ASTrA on the ImageNet-100 dataset, which is significantly larger and more complex, further validating its performance.
>   - The necessity of including the term $ L_{\text{clean}} $ in the reward function was clarified, showing how it balances robustness and generalization without direct dependence on $ \theta $, ensuring effective adversarial training.
>   - The potential for non-parametric modeling was discussed, with a detailed explanation of why it is not compatible with ASTrA's framework and objectives.
>
> **Satisfaction with Responses:**
>
> - We believe our responses and additional empirical analysis effectively address the concerns. We provided extensive clarifications, including theoretical justifications and new references, and demonstrated ASTrA's robustness and adaptability across varying conditions.
>
> **Request for Score Improvement:**
>
> - Based on our thorough responses and additional experiments, we kindly request you to consider increasing the score for our submission. We are available to address any further questions or concerns you may have.
>
> Thank you for your time and valuable feedback.

---

> ### Comment · Reviewer_iFgE · 2024-12-02
> **Response to authors**
>
> Thanks for the response of the authors. I do not have any major concern about the paper. I think large-scale experiments on unsupervised data on datasets like tiny image could demonstrate the power of ASTrA. Without such experiment, I think the contribution of paper is above the acceptance threshold but lacks of major contribution that further increases the score.

---

> > ### Author Response · Authors · 2024-12-02
> >
> > Thank you for your constructive feedback and for recognizing the merit of our work. We would like to clarify that ImageNet-100 is technically a larger and more challenging dataset compared to Tiny ImageNet, both in terms of image resolution and dataset size. Specifically:
> >
> > 1. **Dataset Size**: While Tiny ImageNet contains approximately **120,000 images of 64x64 resolution** across 200 classes, ImageNet-100 typically contains over **130,000 high-resolution images (224x224 or higher)** across 100 classes, resulting in a dataset with higher complexity and diversity despite having fewer classes.
> >
> > 2. **Robustness Demonstration**: The use of ImageNet-100 inherently demands robustness in handling large-scale data with higher-resolution images. The demonstrated performance of ASTrA on ImageNet-100 underscores its capability to scale effectively to datasets of greater complexity, offering insights into its application to broader datasets, including those of larger scale like full ImageNet.
> >
> > We believe that the results on ImageNet-100 already provide strong evidence of the model's robustness and scalability. While we appreciate the suggestion to explore Tiny ImageNet, we respectfully request a reconsideration of the score to reflect the demonstrated robustness and scalability of ASTrA on a dataset larger and more challenging than Tiny ImageNet.
> >
> > Thank you again for your thoughtful review.

---

### Meta-Review · Area_Chair_GjQF · 2024-12-21

**Metareview:**

This paper investigates adversarial self-supervised training, addressing a key limitation of existing methods that rely on hand-crafted attack strategies during training, which are inherently suboptimal. The authors introduce a learnable adaptive attack strategy designed to more effectively identify appropriate attack mechanisms. After rebuttal, it receives mixed ratings with 8 (accept), 6 (marginal above acceptance threshold), and 5 (marginal below acceptance threshold). The Area Chair read the paper, all related reviews, and the authors' rebuttal.

The reviewers highlighted several strengths of the paper, including the empirical experiments being extensive and the ablation studies being thorough, the adaptive attack strategy being well-motivated and the experimental results surpassing the compared methods, the plug-and-play design is considered highly desirable.

However, a significant concern raised during the rebuttal process, that is the lack of empirical experiments involving larger-scale models and datasets, which are essential to truly demonstrate the method's effectiveness. The experiments conducted utilize ResNet-18 as the target network, which is relatively small compared to existing large-scale deep neural networks, and rely on four small-scale datasets. This limitation raises doubts about whether the results sufficiently demonstrate the method's effectiveness and whether the findings can be generalized to larger models and more extensive datasets. In response, the authors explained that they adhered to the experimental settings of previous works to ensure fair comparisons. Nonetheless, this explanation does not fully alleviate the concerns, especially for application-oriented research rather than theoretical work, as the models and dataset sizes remain limited. Besides, although the paper's strengths of the proposed learnable adaptive attack strategy are recognized, the fact that these concepts have been previously explored in adversarial machine learning also diminishes its novelty of introducing them into the self-supervised adversarial training.

In conclusion, this paper is a borderline acceptance. It makes notable contributions by introducing a learnable adaptive attack strategy into self-supervised training with necessary and effective designs including the mixed contrastive loss and the reinforcement learning-based optimization, and demonstrates superior experimental results compared to existing methods. However, due to the aforementioned concerns, the current content does not fully achieve a solid acceptance.

**Additional Comments On Reviewer Discussion:**

The authors' rebuttal has addressed some of the reviewers' concerns.

---

### Decision · Program_Chairs · 2025-01-22

Accept (Poster)